# A dynamic bactofilin cytoskeleton cooperates with an M23 endopeptidase to control bacterial morphogenesis

**Sebastian Pöhl[1†], Manuel Osorio-Valeriano[1,2†‡], Emöke Cserti[1†], Jannik Harberding[1], Rogelio Hernandez-Tamayo[2,3,4], Jacob Biboy[5], Patrick Sobetzko[4§], Waldemar Vollmer[5,6], Peter L Graumann[3,4], Martin Thanbichler[1,2]***

[1]Department of Biology, University of Marburg, Marburg, Germany, Marburg, Germany; [2]Max Planck Institute for Terrestrial Microbiology, Marburg, Germany; [3]Department of Chemistry, University of Marburg, Marburg, Germany; [4]Center for Synthetic Microbiology (SYNMIKRO), Marburg, Germany; [5]Centre for Bacterial Cell Biology, Biosciences Institute, Newcastle University, Newcastle upon Tyne, United Kingdom; [6]Institute for Molecular Bioscience, The University of Queensland, Brisbane, Australia

**\*For correspondence:**
thanbichler@uni-marburg.de

[†]These authors contributed equally to this work

**Present address:** [‡]Department of Cell Biology, Blavatnik Institute, Harvard Medical School, Boston, United States; [§]Faculty of Sciences and Technologies, Université de Lorraine, Nancy, France

**Competing interest:** The authors declare that no competing interests exist.

**Abstract** Bactofilins have emerged as a widespread family of cytoskeletal proteins with important roles in bacterial morphogenesis, but their precise mode of action is still incompletely understood. In this study, we identify the bactofilin cytoskeleton as a key regulator of cell growth in the stalked budding alphaproteobacterium *Hyphomonas neptunium*. We show that, in this species, bactofilin polymers localize dynamically to the stalk base and the bud neck, with their absence leading to unconstrained growth of the stalk and bud compartments, indicating a central role in the spatial regulation of cell wall biosynthesis. Database searches reveal that bactofilin genes are often clustered with genes for cell wall hydrolases of the M23 peptidase family, suggesting a functional connection between these two types of proteins. In support of this notion, we find that the *H. neptunium* M23 peptidase homolog LmdC interacts directly with bactofilin in vitro and is required for proper cell shape in vivo. Complementary studies in the spiral-shaped alphaproteobacterium *Rhodospirillum rubrum* again reveal a close association of its bactofilin and LmdC homologs, which co-localize at the inner curve of the cell, modulating the degree of cell curvature. Collectively, these findings demonstrate that bactofilins and M23 peptidases form a conserved functional module that promotes local changes in the mode of cell wall biosynthesis, thereby driving cell shape determination in morphologically complex bacteria.

## eLife assessment

The manuscript explores the interplay between cytoskeletal bactofilins and cell wall hydrolases in bacterial morphogenesis, utilizing a range of methodologies from bacteriological to biochemical. The study provides **important** insights into the control of peptidoglycan biosynthesis by bactofilin polymers and the function of LdmC, supported by a comprehensive array of genetic, bioinformatic, biochemical, and biophysical tools. These **convincing** findings propose a conserved module governing bacterial morphogenesis, emphasizing the direct association of cell wall remodeling enzymes with a dynamic cytoskeleton, akin to mechanisms observed in other cellular processes such as cell growth and division.

## Introduction

Bacteria come in a variety of different cell shapes, which can be further modified by the formation of cellular extensions such as branches or stalks (*Kysela et al., 2016*; *Yang et al., 2016*). Their morphology can change as a function of the cell cycle or in response to environmental cues to ensure optimal fitness in the given ecological niche or growth conditions (*van Teeseling et al., 2017*; *Yang et al., 2016*). In the vast majority of species, cell shape is determined by the peptidoglycan cell wall, a complex macromolecule composed of glycan chains that are crosslinked by short peptides (*Typas et al., 2012*; *Vollmer et al., 2008*). The synthesis of this mesh-like structure is achieved by an array of synthetic and lytic enzymes that are typically combined into multi-protein complexes and associated with regulatory factors and cytoskeletal elements to facilitate the coordination and spatiotemporal regulation of their activities (*Egan et al., 2020*; *Rohs and Bernhardt, 2021*).

Basic spherical, rod-like, and hyphal shapes are generated by the combined action of the cell division and cell elongation machinery (*Margolin, 2009*; *Rohs and Bernhardt, 2021*). In most bacteria, cell division is executed by the divisome, which is organized by the tubulin homolog FtsZ and mediates cell constriction and the synthesis of the new cell poles prior to cytokinesis (*McQuillen and Xiao, 2020*). Cell elongation, by contrast, can be achieved by various types of cell wall-biosynthetic complexes, including the so-called elongasome, which is organized by the actin homolog MreB and mediates the dispersed incorporation of new cell wall material along the lateral cell walls (*Shi et al., 2018*), or different cell pole-associated complexes that promote polar growth of the cell body (*Brown et al., 2011*). More complex cell shapes are generated with the help of accessory systems that either modulate the activity of the generic cell elongation machinery or have peptidoglycan biosynthetic activity on their own, thereby locally modifying the structure of the peptidoglycan layer (*Taylor et al., 2019*).

A particularly widespread family of cytoskeletal proteins implicated in cell shape modification are the bactofilins. They are characterized by a conserved central Bactofilin A/B domain (InterPro ID: IPR007607; *Paysan-Lafosse et al., 2023*) with a barrel-like β-helical fold that is typically flanked by short disordered terminal regions (*Kühn et al., 2010*; *Shi et al., 2015*; *Vasa et al., 2015*). Bactofilins polymerize spontaneously without the need for nucleotide cofactors (*Koch et al., 2011*; *Kühn et al., 2010*), driven by head-to-head and tail-to-tail interactions between the core domains of neighboring molecules (*Deng et al., 2019*). Lateral interactions between individual protofilaments can then give rise to higher-order assemblies, such as bundles or two-dimensional sheets (*Kühn et al., 2010*; *Vasa et al., 2015*; *Zuckerman et al., 2015*). Previous work has suggested that bactofilin polymers typically associate with the inner face of the cytoplasmic membrane and localize to regions of high membrane curvature (*Caccamo et al., 2020*; *Hay et al., 1999*; *Kühn et al., 2010*; *Lin et al., 2017*; *Taylor et al., 2020*). These structures have been co-opted as localization determinants and assembly platforms by several different morphogenetic systems.

For instance, bactofilin homologs were reported to contribute to rod-shape maintenance in *Myxococcus xanthus* (*Koch et al., 2011*), the establishment of helical cell shape in the human pathogen *Helicobacter pylori* (*Sycuro et al., 2010*; *Taylor et al., 2020*) as well as the modulation of cell helicity in the spiral-shaped bacterium *Leptospira biflexa* (*Jackson et al., 2018*). Apart from modulating general cell shape, they were found to have an important role in the formation of cellular extensions known as stalks, which are widespread among alphaproteobacterial species (*Wagner and Brun, 2007*). Stalks are elongated protrusions of the cell envelope that are filled with a thin thread of cytoplasm and grow through zonal incorporation of cell wall material at their base (*Aaron et al., 2007*; *Randich and Brun, 2015*). In *Caulobacter crescentus* and *Asticcacaulis biprosthecum*, bactofilin polymers were shown to localize to the stalk base to direct proper stalk formation. In *C. crescentus*, they recruit a cell wall synthase that contributes to stalk elongation, with their absence leading to a reduction in stalk length (*Kühn et al., 2010*). In *A. biprosthecum*, by contrast, bactofilin acts as a central topological regulator that is required to efficiently initiate stalk formation and limit peptidoglycan biosynthesis to the stalk base. Its absence leads to the development of pseudostalks, which are much shorter and wider than normal stalks and irregularly shaped, likely due to unrestrained peptidoglycan biosynthesis through the entire stalk envelope (*Caccamo et al., 2020*).

While stalks are often accessory structures with highly specialized functions (*Klein et al., 2013*; *Persat et al., 2014*; *Wagner and Brun, 2007*), stalked budding bacteria such as *Hyphomonas neptunium* and other members of the *Hyphomonadaceae* and *Hyphomicrobiaceae* use them as integral

parts of the cell with key roles in cell growth and division (*Moore, 1981*). *H. neptunium* has a biphasic life cycle (*Wali et al., 1980*), in which a non-replicative, motile swimmer cell sheds its single polar flagellum and differentiates into a replicative, sessile stalked cell (*Figure 1—figure supplement 1*). Unlike most widely studied model species, it does not divide by binary fission but instead produces new offspring through the formation of buds at the tip of the stalk. As the terminal stalk segment gradually dilates, a flagellum is formed at the pole opposite the stalk. After DNA replication and trans-location of one of the sister chromosomes through the stalk into the bud compartment (*Jung et al., 2019*), cytokinesis occurs at the bud neck, releasing a new swimmer cell. While the newborn cell first needs to differentiate into stalked cells to start replication, the stalked mother cell restores the stalk and then immediately re-enters the next budding cycle (*Jung et al., 2019*; *Wali et al., 1980*). The developmental program of *H. neptunium* involves several switches in the pattern of peptidoglycan biosynthesis (*Cserti et al., 2017*). After birth, swimmer cells increase in size by dispersed incorporation of new cell wall material throughout the cell body. Stalk formation is then achieved by zonal growth at the stalk base, followed by localized dispersed growth of the stalk-terminal bud compartment and, finally, zonal peptidoglycan synthesis at the site of cell division. The pattern of new cell wall synthesis is similar to the localization pattern of elongasome components (*Cserti et al., 2017*), suggesting the involvement of this machinery in all growth phases. However, the underlying regulatory mechanisms are still unknown.

In this study, we identify the bactofilin cytoskeleton as a central player in the regulation of cell growth of *H. neptunium*. We show that the lack of bactofilins causes severe morphological defects, resulting from unconstrained growth of the stalk and bud compartments. In line with this finding, bactofilin polymers localize dynamically to the stalk base and then to the incipient bud neck prior to the onset of bud formation, acting as a barrier that retains the cell wall biosynthetic machinery in the respective growth zones. Notably, in a broad range of species, bactofilin genes lie adjacent to genes encoding putative M23 peptidases. We show that the corresponding *H. neptunium* homolog, LmdC, is an essential bitopic membrane protein with peptidoglycan hydrolase activity that interacts directly with bactofilin through its N-terminal cytoplasmic tail. Its CRISPRi-mediated depletion also results in unconstrained cell growth, confirming its functional connection to bactofilin. The conservation of this interaction is further verified by studies of the spiral-shaped alphaproteobacterium *Rhodospirillum rubrum*. We show that the bactofilin and LmdC homologs of this species co-localize at the inner curve of the cell, forming filamentous assemblies that modulate the degree of cell curvature. Their interaction is abolished by amino acid exchanges in the cytoplasmic tail of LmdC, confirming a key role of this region in the recruitment of LmdC. Together, these results demonstrate a conserved functional interaction between bactofilins and M23 peptidases that is important for the control of cell growth in morphologically complex bacteria.

## Results

### The bactofilin cytoskeleton is required for cell shape determination in *H. neptunium*

The *H. neptunium* chromosome contains two open reading frames that encode bactofilin homologs, HNE_0444 and HNE_2629 (*Badger et al., 2006*). Reciprocal BLAST searches identified HNE_2629 as a potential ortholog of *C. crescentus* BacA (43% identity, 60% similarity), whereas HNE_0444 was only distantly related (<30% identity) to other so-far characterized bactofilin homologs. Based on these results, we propose to designate the two bactofilin homologs of *H. neptunium* BacA (HNE_2629) and BacD (HNE_0444), respectively. *bacA* forms a putative bicistronic operon with *lmdC*, an essential gene encoding an M23 peptidase homolog (*Cserti et al., 2017*; *Figure 1A*). The two genes overlap by 17 base pairs, suggesting that their expression is closely coupled. *bacD*, by contrast, is not part of an operon. Both BacA and BacD display the typical architecture of bactofilins, with a central polymerization domain flanked by short non-structured N- and C-terminal regions (*Figure 1B*).

To investigate the role of bactofilins in *H. neptunium*, we generated mutant strains in which *bacA* and *bacD* were deleted either individually or in combination and analyzed their phenotypes (*Figure 1*). The Δ*bacA* and Δ*bacAD* cells showed severe morphological defects, as reflected by irregularly shaped, elongated and/or oversized cells, buds directly fused with the mother cell body, branched stalks, and multiple wide protrusions that emerged from the cells in an apparently random fashion (*Figure 1C–E*),

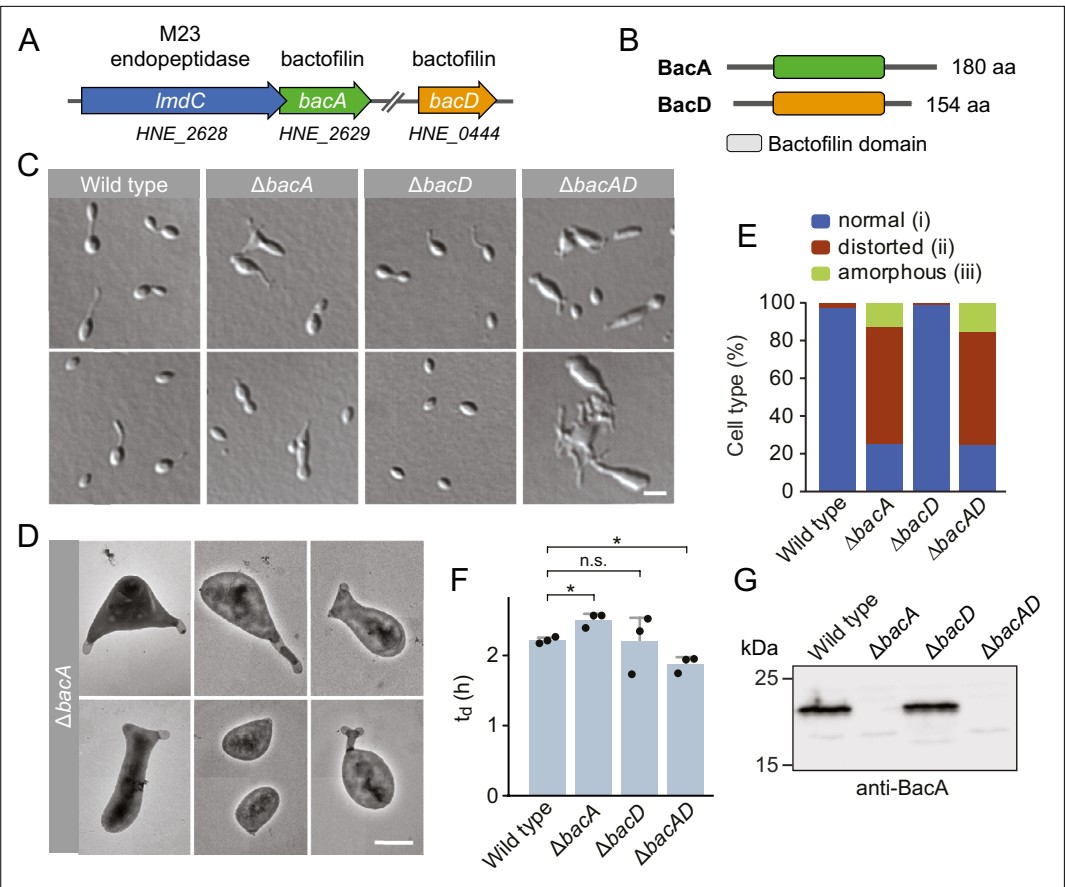

**Figure 1.** The bactofilin homolog BacA is required for proper cell morphology in *H. neptunium*. (**A**) Schematic representation of the two bactofilin genes present in the *H. neptunium* genome. *bacA* lies adjacent to the M23 peptidase gene *lmdC*. Arrows indicate the direction of transcription. (**B**) Domain organization of BacA and BacD from *H. neptunium*. The bactofilin polymerization domain (colored boxes) is flanked by non-structured N- and C-terminal regions. (**C**) Morphology of *H. neptunium* bactofilin mutants. Shown are representative cells of strains LE670 (wild type), EC28 (Δ*bacA*), EC23 (Δ*bacD*), and EC33 (Δ*bacAD*), imaged by differential interference contrast (DIC) microscopy. Bar: 4 µm. (**D**) Transmission electron micrographs of representative Δ*bacA* cells at the early stalked-cell stage. Bar: 1 µm. (**E**) Quantification of the proportion of phenotypically abnormal stalked and budding cells in cultures of the strains analyzed in panel C (n=100 cells per strain). (**F**) Doubling times of the indicated *H. neptunium* strains. Bars represent the mean (± SD) of three independent experiments (dots). Statistical significance was determined using an unpaired two-sided Welch's t-test (*p<0.05; n.s., not significant). (**G**) Immunoblot analysis of the strains shown in panel C, performed using anti-BacA antibodies.

The online version of this article includes the following source data and figure supplement(s) for figure 1:

**Source data 1.** Raw images for the immunoblot analysis in *Figure 1G*.

**Figure supplement 1.** The dimorphic lifecycle of *H. neptunium*.

**Figure supplement 2.** Phenotypic analysis of *H.neptunium* bactofilin mutants.

**Figure supplement 2—source data 1.** Raw images for the immunoblot analysis in *Figure 1—figure supplement 2A*.

reminiscent of the pseudostalks reported for a bactofilin-deficient *A. biprosthecum* mutant (*Caccamo et al., 2020*). Surprisingly, these growth defects did not have any major defects on the doubling time (*Figure 1F*). The wild-type phenotype could be largely restored by expressing an ectopic copy of *bacA* under the control of a copper-inducible promoter, even though BacA accumulated to lower-than-normal levels under this condition, confirming the absence of polar effects (*Figure 1—figure supplement 2A–D*). The deletion of *bacD*, by contrast, did not cause any obvious cell shape defects (*Figure 1C, E and F* and *Figure 1—figure supplement 2E*). Moreover, neither *bacD* deletion nor *bacD* overexpression had any influence on the proportion of distorted or amorphous cells in the

Δ*bacA* background (*Figure 1C and F* and *Figure 1—figure supplement 2C and F*). These results demonstrate that BacA has a critical role in the regulation of cell growth in *H. neptunium*, whereas BacD might be an auxiliary factor of so-far unknown function, similar to BacB in *C. crescentus* (*Kühn et al., 2010*).

To gain more insight into the dynamics of cell growth in the Δ*bacA* mutant and identify the initial phenotypic defects induced upon BacA depletion, we imaged a conditional *bacA* mutant after its transfer from permissive to restrictive conditions on an agarose pad (*Figure 2A and B*). Following cells at the swimmer-to-stalked cell transition, we observed that stalk formation initially proceeded as in the wild-type strain. However, as BacA was gradually depleted, stalk elongation ceased and the stalk structure started to widen and eventually develop multiple bulges that kept on expanding in an apparently uncontrolled manner. Thus, BacA appears to be required to maintain the polar growth zone at the stalk base, with its absence leading to unconstrained growth of the stalk cell wall. To follow the fate of bactofilin-deficient cells over a prolonged period of time, we monitored Δ*bacAD* cells in a flow-cell system, which ensured optimal nutrient supply throughout the course of the experiment and led to a looser packing of cells, thereby facilitating their visual analysis (*Figure 2—figure supplement 1* and *Figure 2—video 1*). In this setup, cells again started growth by the formation of irregularly shaped stalks that started to branch, with branches developing either into extensive hyphal-like structures or large, amorphous compartments that may represent morphologically aberrant buds. At irregular intervals, cells divided at the junctions between hyphal and bulged cellular segments, releasing smaller amorphous fragments and, occasionally, also cells with close-to-normal morphology. Wild-type cells, by contrast, showed the usual growth pattern when cultivated under these conditions (*Figure 2—video 2*). Together, these results underscore the importance of BacA in the spatiotemporal control of cell growth in *H. neptunium*.

To obtain more detailed information about the dynamics of cell wall biosynthesis in the absence of bactofilins, we performed pulse-labeling studies with the fluorescent D-amino acid HADA, which is rapidly incorporated into peptidoglycan when added to the culture medium, thus marking regions of ongoing cell wall biosynthesis (*Kuru et al., 2015*). Wild-type cells showed the typical succession of growth modes, with dispersed growth in swimmer cells, zonal growth at the stalk base in stalked cells, and localized dispersed growth in the nascent bud (*Figure 2C*). In the Δ*bacAD* background, by contrast, this switch in the growth modes was abolished. Cells with close-to-normal morphology that had just initiated stalk formation still showed a distinct fluorescent focus at the stalked pole, suggesting that the initial recruitment of the machinery responsible for stalk formation occurred in a bactofilin-independent manner. However, all other cell types, including amorphous cells with aberrant stalk- and bud-like extensions, only displayed diffuse fluorescence, which points to uncontrolled growth by dispersed incorporation of new peptidoglycan throughout the entire cell envelope. These findings support the notion that the bactofilin cytoskeleton is required to limit cell wall biosynthesis to the different growth zones of *H. neptunium*.

## The *H. neptunium* bactofilin homologs assemble into stable filamentous structures

Despite their severe cell shape defects, the Δ*bacA* and Δ*bacAD* mutants showed close-to-normal growth rates. Moreover, muropeptide analysis revealed that the global composition of their peptidoglycan remained essentially unchanged (*Figure 2—figure supplement 2* and *Supplementary file 1*). These findings suggested that BacA might affect cell wall composition at a local scale or act mainly by modulating the activity of the generic cell elongation machinery. To further investigate the role of bactofilins in *H. neptunium*, we first aimed to verify the ability of the proteins to assemble into polymeric scaffolds. A model of the structure of BacA generated with AlphaFold-Multimer (*Evans et al., 2022*) suggested that the protein adopted a β-helical fold and polymerized through head-to-head and tail-to-tail interactions between different subunits (*Figure 3A*), as revealed previously in experimental studies (*Deng et al., 2019*; *Shi et al., 2015*). Consistent with this prediction, purified BacA produced a mixture of long filaments, filament bundles and two-dimensional polymeric sheets in vitro, which could be readily visualized by transmission electron microscopy (*Figure 3B*). Moreover, upon heterologous co-expression in *Escherichia coli*, fluorescently tagged derivatives of BacA and BacD co-localized into extended filamentous structures that were associated with the cell envelope (*Figure 3C* and *Figure 3—figure supplement 1A*). The BacA fusion formed qualitatively similar

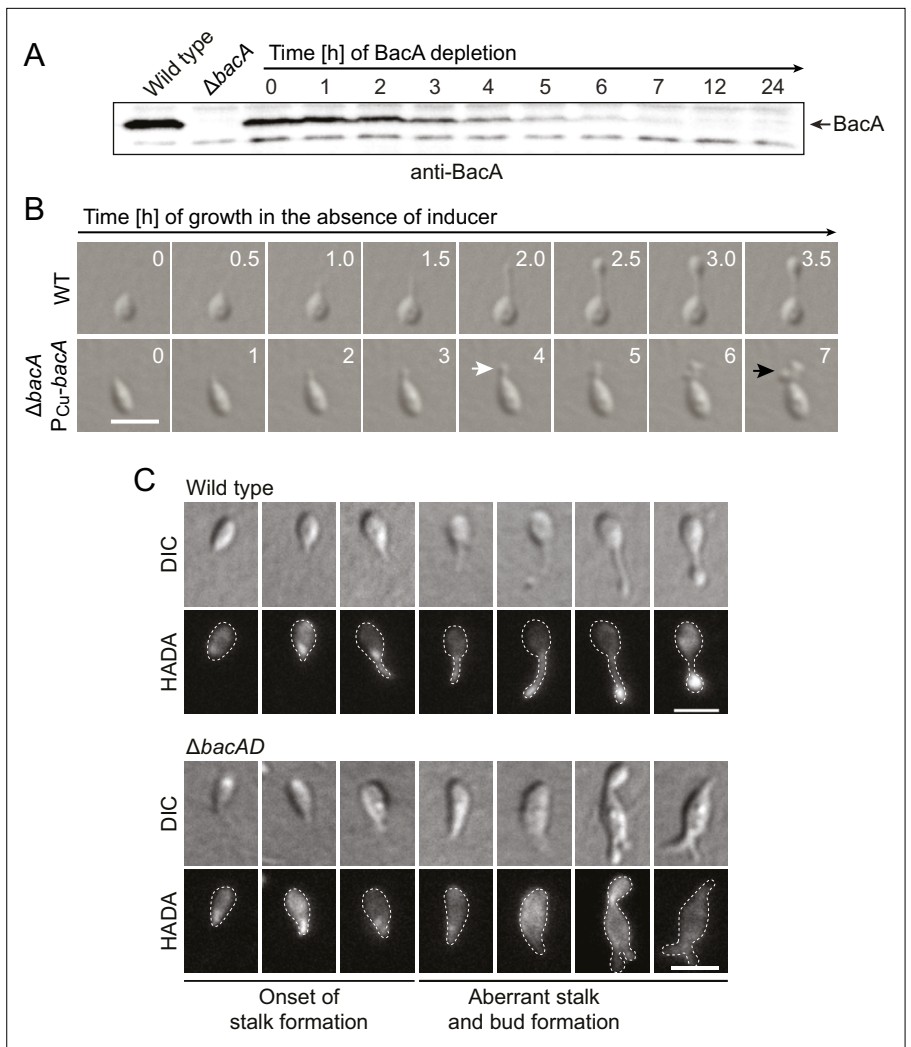

**Figure 2.** Lack of BacA leads to uncontrolled growth of the stalk and bud compartments. (**A**) Immunoblot showing the levels of BacA in strain EC41 ($\Delta bacA$ $P_{Cu}$-$bacA$) over the course of BacA depletion. Cells were grown in copper-containing medium, washed, and transferred to inducer-free medium. At the indicated time points, samples were withdrawn and subjected to immunoblot analysis. Strains LE670 (wild-type) and EC28 ($\Delta bacA$) were included as controls. The position of BacA is indicated. (**B**) Morphological defects induced by BacA depletion. Cells of strains LE670 (wild-type) and EC41 ($\Delta bacA$ $P_{Cu}$-$bacA$) were grown in medium containing 0.5 mM $CuSO_4$, washed, and incubated for 6 hr in inducer-free medium before they were transferred onto ASS-agarose pads lacking inducer and imaged at the indicated time points. Shown are representative images. Bar: 2 µm. (**C**) Changes in the growth pattern of *H. neptunium* in the absence of bactofilins. Cells of strains LE670 (wild-type), and EC33 ($\Delta bacAD$) were stained with the fluorescent D-amino acid HADA prior to analysis by fluorescence microscopy. Shown are representative images of cells at different developmental stages. Bars: 2 µm.

The online version of this article includes the following video, source data, and figure supplement(s) for figure 2:

**Source data 1.** Raw images for the immunoblot analysis in *Figure 2A*.

**Figure supplement 1.** Growth of $\Delta bacAD$ cells in a microfluidic flow cell.

**Figure supplement 2.** Muropeptide profiles of different *H. neptunium* strains.

**Figure 2—video 1.** Unconstrained growth of a bactofilin-deficient *H. neptunium* mutant.
https://elifesciences.org/articles/86577/figures#fig2video1

**Figure 2—video 2.** Normal growth of the *H. neptunium* wild-type strain.
https://elifesciences.org/articles/86577/figures#fig2video2

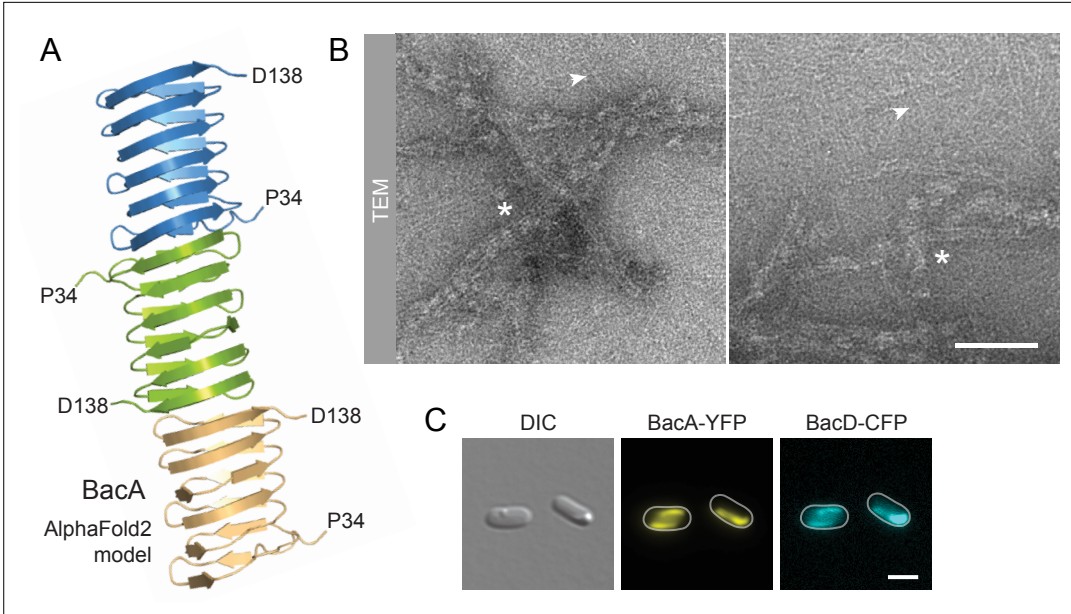

**Figure 3.** BacA and BacD assemble into filamentous structures. (**A**) Model of a BacA trimer generated with AlphaFold-Multimer (*Evans et al., 2022*). Only residues P34-D138 are shown for each subunit. (**B**) Visualization of BacA polymers. Purified BacA-His₆ was stained with uranyl acetate and imaged by transmission electron microscopy (TEM). Arrowheads point to BacA filaments. Asterisks indicate filament bundles and sheets. Bars: 200 nm. (**C**) Copolymerization of BacA and BacD after heterologous co-expression in *E. coli*. Cells of *E. coli* BL21(DE3) transformed with plasmid pEC121 (P$_{T7}$-*bacA-eyfp* P$_{T7}$-*bacD-ecfp*) were grown in LB medium containing 5% glucose and induced with 0.5 mM isopropyl-β-D-1-thiogalactopyranoside (IPTG) prior to imaging. Shown are representative cells. Bar: 3 μm.

The online version of this article includes the following figure supplement(s) for figure 3:

**Figure supplement 1.** Localization patterns of BacA-YFP and BacD-CFP after heterologous overproduction in *E. coli*.

---

structures when produced alone, whereas the BacD fusion condensed into tight foci in the absence of BacA (*Figure 3—figure supplement 1B and C*). These results confirm the polymeric nature of bactofilin assemblies in *H. neptunium* and suggest that BacA and BacD interact with each other in vivo.

## Bactofilins localize dynamically to the boundaries of the *H. neptunium* growth zones

Next, we sought to investigate the localization dynamics of the bactofilin cytoskeleton in vivo. To this end, we generated strains producing fluorescently tagged BacA or BacD derivatives in place of the native proteins (*Figure 4—figure supplement 1*). Time-lapse microscopy analysis of cells producing a BacA-YFP fusion on agarose pads (*Figure 4A*) revealed that the protein was localized to the new cell pole in swimmer cells and remained associated with the stalk base during the initial phase of stalk formation. At some point, it appeared to attach to the stalk structure and then move away from the base as new cell wall material was inserted. Subsequently, stalk elongation ceased and the terminal stalk segment, delimited by the bactofilin assembly, started to swell and develop into a bud, gradually displacing BacA-YFP in the direction of the stalk base as its size increased. After cell division, both the mother cell and the newborn swimmer cells showed a fluorescent focus, indicating that the bactofilin assembly was split during cell division. To monitor the dynamics of the bactofilin cytoskeleton over multiple division cycles, we analyzed the same strain in a flow-cell system (*Figure 4—figure supplement 2* and *Figure 4—videos 1–3*). Under these conditions, the small, newborn swimmer cells were washed away immediately after cytokinesis, preventing the formation of microcolonies around the mother cell. We observed that cell division occurred at a small distance from the BacA-YFP focus, leaving a short stalk-terminal segment in between the bactofilin assembly and the stalk tip. After cytokinesis, the stalk elongated again prior to the initiation of the next budding event. Notably, BacA-YFP

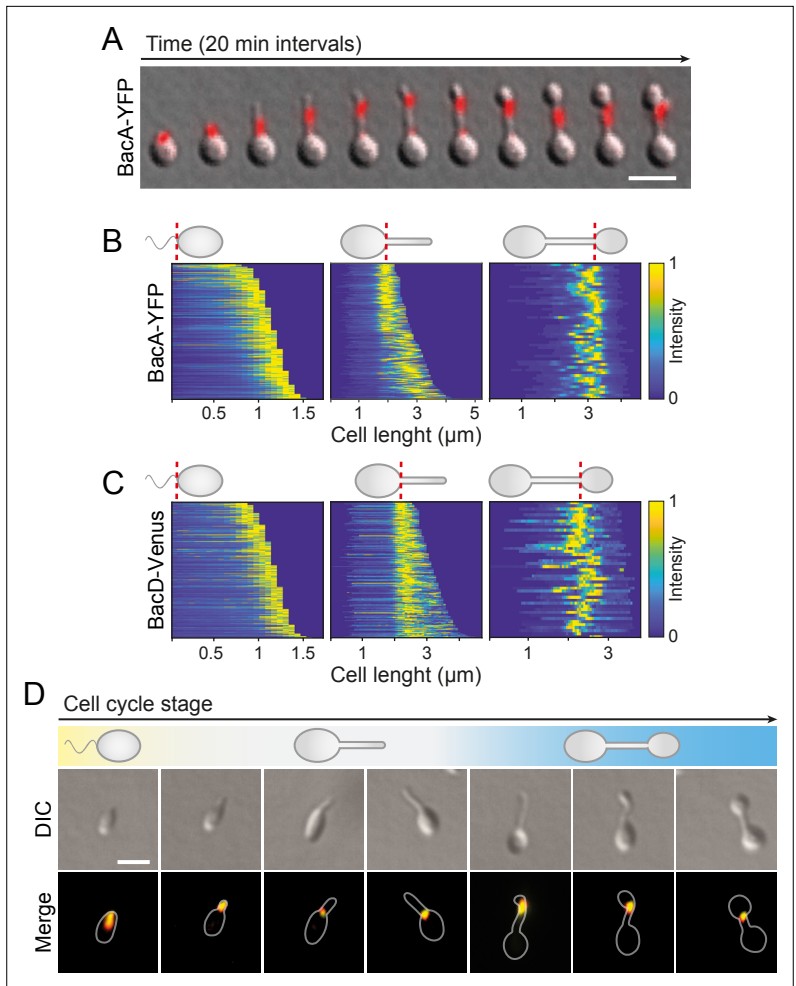

**Figure 4.** BacA and BacD show a dynamic, cell cycle-dependent localization pattern. (**A**) Localization pattern of BacA-YFP. Cells of strain EC61 (*bacA::bacA-eyfp*) were transferred to agarose pads and imaged at 20 min intervals. Shown are overlays of DIC and fluorescence images of a representative cell, with YFP fluorescence shown in red. Scale bar: 3 µm. (**B**) Demographic analysis of BacA-YFP localization in swimmer (left), stalked (middle), and budding (right) cells of strain EC61 (*bacA::bacA-eyfp*). The fluorescence intensity profiles measured for cells of each type were sorted according to cell length and stacked on each other, with the shortest cell shown at the top and the longest cell shown at the bottom (n=250 swimmer cells, 215 stalked cells, and 49 budding cells). Dotted red lines indicate the positions to which cells were aligned. (**C**) Demographic analysis of BacD-Venus localization. Cells of strain EC67 (*bacD::bacD-venus*) were analyzed as described in panel B (n=416 swimmer cells, 248 stalked cells, and 45 budding cells). (**D**) Co-localization of BacA and BacD in cells of strain EC68 (*bacA::bacA-eyfp bacD::bacD-mCherry*). Shown are DIC images and overlays of the YFP and mCherry signals of representative cells, arranged according to their developmental state. The individual signals are shown in *Figure 6A*. The Pearson's Correlation Coefficient (PCC) of the two fluorescence signals in a random population of cells (n=454) is 0.506. Bar: 1 µm.

The online version of this article includes the following video, source data, and figure supplement(s) for figure 4:

**Figure supplement 1.** Stability of the fluorescently tagged bactofilin derivatives used in this study.

**Figure supplement 1—source data 1.** Raw images for the immunoblot analyses in *Figure 4—figure supplement 1A and B*.

**Figure supplement 2.** Localization dynamics of BacA-YFP.

**Figure supplement 3.** Co-localization of BacA and BacD.

**Figure supplement 4.** Partial independence of BacA and BacD localization at the onset of the budding process.

**Figure supplement 5.** Random localization of BacD complexes in the absence of BacA.

**Figure supplement 6.** Localization and functionality of BacA$_{F130R}$-YFP.

**Figure 4—video 1.** Localization dynamics of BacA-YFP (example 1).

*Figure 4 continued on next page*

*Figure 4 continued*

https://elifesciences.org/articles/86577/figures#fig4video1

**Figure 4—video 2.** Localization dynamics of BacA-YFP (example 2).

https://elifesciences.org/articles/86577/figures#fig4video2

**Figure 4—video 3.** Localization dynamics of BacA-YFP (example 3).

https://elifesciences.org/articles/86577/figures#fig4video3

was only occasionally detected at the stalked pole during the stalk restoration phase, suggesting that the bactofilin cytoskeleton no longer plays a major role in stalk growth once the stalk structure has been established.

To verify the behavior observed in the time-lapse studies, we performed a population-wide analysis of the BacA-YFP localization pattern, based on snap-shot images of exponentially growing cells. A demographic analysis confirmed that the protein localizes to the new cell pole in swimmer cells, remains at the stalk base in cells with short stalks, and then moves to a position close to the distal end of the stalk before the onset of budding, remaining associated with the bud neck up to the point of cell division (*Figure 4B*). A very similar behavior was observed for a BacD variant fused to the yellow fluorescent protein Venus (*Nagai et al., 2002*; *Figure 4C*), consistent with the notion that the two bactofilin paralogs interact. In support of this hypothesis, studies of a strain in which both proteins were fluorescently labeled showed that BacA and BacD indeed co-localized to a large extent during all stages of the developmental cycle (*Figure 4D* and *Figure 4—figure supplement 3*). However, at the onset of bud formation, the signals of the two proteins were not always fully superimposable, with BacD assemblies occasionally detectable both at the bud neck and in the mother cell (*Figure 4—figure supplement 4*). In the absence of BacA, BacD-Venus still formed distinct foci, albeit at apparently random positions within the cell (*Figure 4—figure supplement 5*). Notably, a BacA-YFP variant carrying a previously reported mutation that disrupts the polymerization interface (F130R) (*Deng et al., 2019*; *Vasa et al., 2015*; *Zuckerman et al., 2015*) was evenly distributed in the cell and unable to functionally replace the wild-type protein, indicating that the formation of polymeric assemblies is essential for proper BacA localization and function (*Figure 4—figure supplement 6*). Given that the bactofilin assemblies are localized to the boundaries of the stalk and the bud and required to prevent unconstrained growth of these cellular structures, we hypothesized that the bactofilin cytoskeleton might serve to limit the cell wall biosynthetic machinery to the different growth zones of *H. neptunium*.

To further investigate the role of bactofilin in the spatial regulation of cell wall biosynthesis, we generated derivatives of the wild-type and Δ*bacAD* strains that produced a fluorescently (YFP-) labeled variant of the membrane-bound elongasome protein RodZ, a core structural component of the complex connecting the cytoplasmic MreB cytoskeleton to the periplasmic enzymatic machinery (*Bendezú et al., 2009*; *van den Ent et al., 2010*; *Alyahya et al., 2009*). In the wild-type background, YFP-RodZ formed dynamic foci in the mother cell, at the (incipient) stalk base and the nascent bud (*Figure 5* and *Figure 5—videos 1 and 2*), thus localizing to the different growth zones detected by HADA labeling (*Figure 2C*). These results are in line with previous findings identifying the elongasome as a key determinant of cell growth in *H. neptunium* (*Cserti et al., 2017*). Importantly, however, YFP-RodZ foci were never observed in the stalk, suggesting that elongasome complexes are unable to cross the stalk base and the bud neck, even though their free components appear to be able to diffuse through the stalk to facilitate the de novo assembly of elongasomes in the incipient bud compartment. In the Δ*bacAD* mutant, the localization behavior of YFP-RodZ resembled that in the wild-type background as long as the cells were the early stages of the cell cycle, with dynamic foci detectable throughout the cell envelope and at the (future) stalk base (*Figure 5* and *Figure 5—video 3*). However, at later stages, the elongasome was no longer excluded from the nascent stalk structure and, in many cases, even appeared to be enriched this region, leading to its remodeling into an increasingly amorphous cellular extension (*Figure 5* and *Figure 5—videos 3–5*). The bactofilin cytoskeleton itself or the high degree of positive inner cell curvature induced in its presence at the stalk base thus appear to act as a barrier that excludes elongasome complexes from the stalk and thus facilitates proper morphogenesis and bud development.

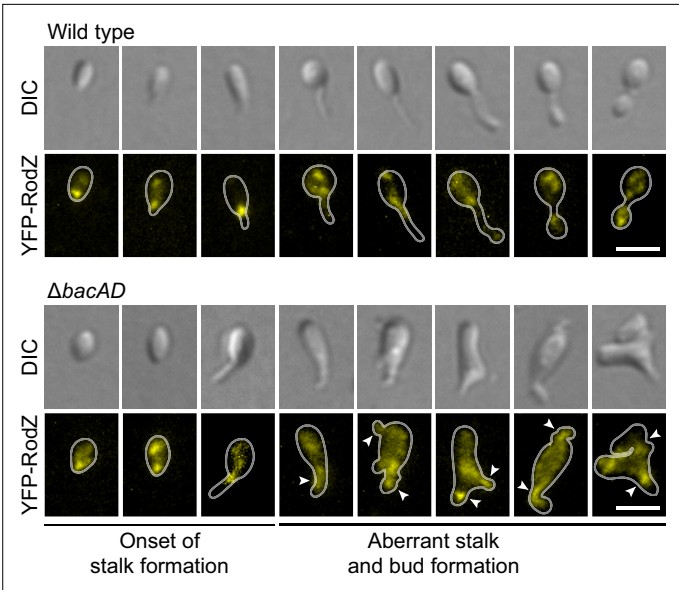

**Figure 5.** The bactofilin cytoskeleton is required to confine the elongasome to the mother cell and bud compartments. Localization of YFP-RodZ in the *H. neptunium* wild-type and Δ*bacAD* backgrounds. Shown are representative cells of strain EC93 (*rodZ::eyfp-rodZ*) and SP221 (Δ*bacAD rodZ::eyfp-rodZ*) imaged by DIC and fluorescence microscopy and arranged according to their developmental state. Arrowheads indicate cellular extensions that show YFP-RodZ foci. Bar: 2 μm.

The online version of this article includes the following video(s) for figure 5:

**Figure 5—video 1.** Localization of YFP-RodZ during stalk synthesis in the wild-type background.
https://elifesciences.org/articles/86577/figures#fig5video1

**Figure 5—video 2.** Localization of YFP-RodZ during bud formation in the wild-type background.
https://elifesciences.org/articles/86577/figures#fig5video2

**Figure 5—video 3.** Localization of YFP-RodZ during pseudo stalk synthesis in the Δ*bacAD* background.
https://elifesciences.org/articles/86577/figures#fig5video3

**Figure 5—video 4.** Localization of YFP-RodZ within early stalks in the Δ*bacAD* background.
https://elifesciences.org/articles/86577/figures#fig5video4

**Figure 5—video 5.** Localization of YFP-RodZ in amorphous Δ*bacAD* cells.
https://elifesciences.org/articles/86577/figures#fig5video5

## The assembly state of BacA changes at different stages of *H. neptunium* development

Since bactofilins form highly stable polymers in vitro (*Kühn et al., 2010*; *Zuckerman et al., 2015*), the dynamic localization observed for BacA and BacD was unexpected. To further characterize the dynamics of the bactofilin cytoskeleton in *H. neptunium*, we followed the movement of individual BacA-YFP molecules in swimmer, stalked, and budding cells (*Figure 6—video 1*). First, we used single-molecule tracking data to generate high-resolution images of the bactofilin assemblies in each of the three cell types. The results confirmed the cell cycle-dependent localization patterns observed by widefield fluorescence microscopy (*Figure 6A* and *Figure 6—figure supplement 1*). In addition, they revealed clusters in medial regions of the stalk, which may represent transient polymers formed during the reshuffling of bactofilin molecules between the stalk base and the (incipient) bud neck. When calculating the average mean squared displacement of the tracked BacA-YFP molecules, we found that their mobility decreased gradually from the swimmer over the stalked cell to the budding cell stage (*Figure 6B*). For each cell type, the distribution of step sizes in the single-particle tracks suggests the existence of two distinct diffusion regimes, with a static (D=0.02 ± 0.0004 μm² s⁻¹) and a mobile (D=0.35 ± 0.004 μm² s⁻¹) population, likely representing the polymerized and freely diffusible states, respectively (*Figure 6C* and *Figure 6—figure supplement 2*). In swimmer cells, the static fraction comprised only ~60% of the molecules. Its proportion increased to ~70% in stalked cells and

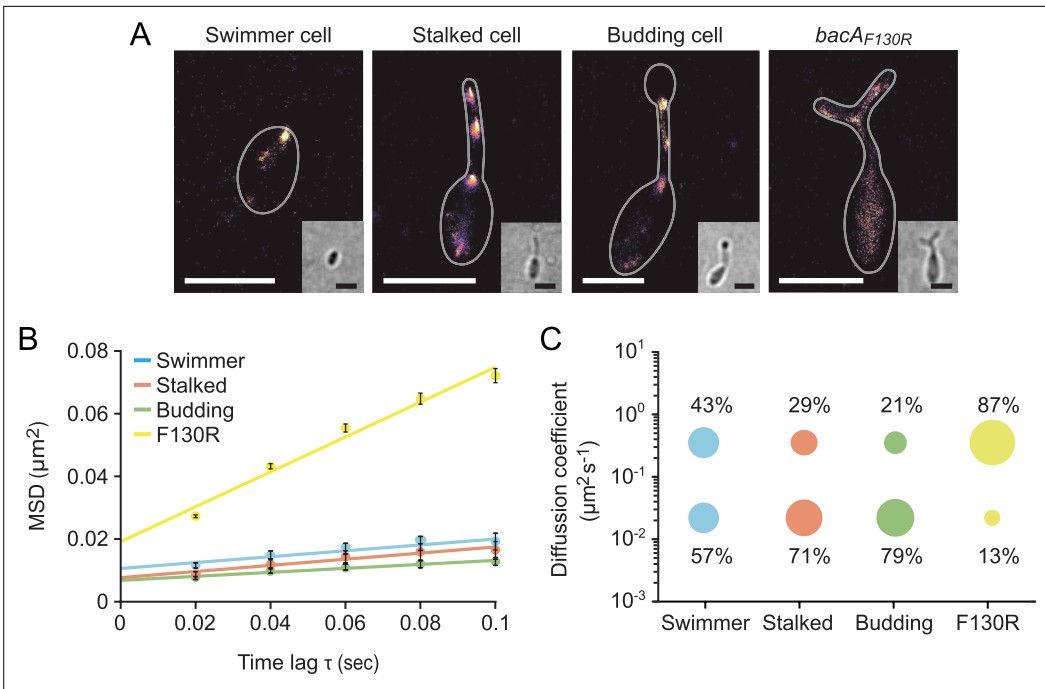

**Figure 6.** The single-particle dynamics of BacA change over the course of the cell cycle. (**A**) Representative heat maps showing the sum of single-particle positions observed in live-cell single-particle localization microscopy studies of a swimmer, stalked, and budding cell producing BacA-YFP (EC61) or a mutant cell producing the polymerization-deficient BacA$_{F130R}$-YFP variant (MO78). Insets show the corresponding bright-field images. Bars: 1 µm. (**B**) Mean-squared displacement (MSD) analysis of the mobility of BacA-YFP in wild-type swimmer (n=100), stalked (n=105), and budding (n=101) cells (EC61). Cells (n=110) producing the BacA$_{F130R}$-YFP were analyzed as a control (MO78). Error bars indicate the standard deviation. (**C**) Bubble plots showing the proportions of mobile and immobile particles (size of the bubbles) and the corresponding average diffusion constants (y-axis) in the cells analyzed in panel B.

The online version of this article includes the following video and figure supplement(s) for figure 6:

**Figure supplement 1.** Single-particle tracking analysis of BacA-YFP.

**Figure supplement 2.** Single-particle tracking analysis of BacA-YFP mobility.

**Figure 6—video 1.** Tracking of single BacA-YFP particles in representative cells at different stages of the cell cycle. https://elifesciences.org/articles/86577/figures#fig6video1

---

finally reached ~80% in budding cells. The F130R variant, by contrast, showed less than 15% of static molecules (*Figure 6B and C* and *Figure 6—figure supplement 2* and *Figure 6—video 1*) consistent with the notion that it is impaired in polymerization but still able to assemble to some extent after undergoing a structural change that generates an alternative polymerization interface (*Deng et al., 2019*). Collectively, these results show that, in *H. neptunium*, the assembly state of the bactofilin cytoskeleton changes as cells progress through their developmental cycle. Moreover, despite the inherent stability of bactofilin polymers, all cell types display a sizeable fraction of mobile BacA-YFP molecules, which could potentially reflect the dynamic reorganization of bactofilin assemblies during cell growth.

## Bactofilin genes are often clustered with genes encoding M23 peptidases

Having identified a critical role of BacA in *H. neptunium* morphogenesis, we set out to gain further insight into its mechanism of action. Notably, in several species in which bactofilins play a role in cell shape determination, the corresponding bactofilin genes are located adjacent to genes encoding putative M23 peptidases homologous to LmdC (*Figure 7A*). To determine whether this gene arrangement was more widely conserved, we performed a comprehensive bioinformatic analysis in which we searched all bacterial genome sequences available for putative bactofilin genes that were located immediately upstream or downstream of open reading frames encoding proteins with a predicted

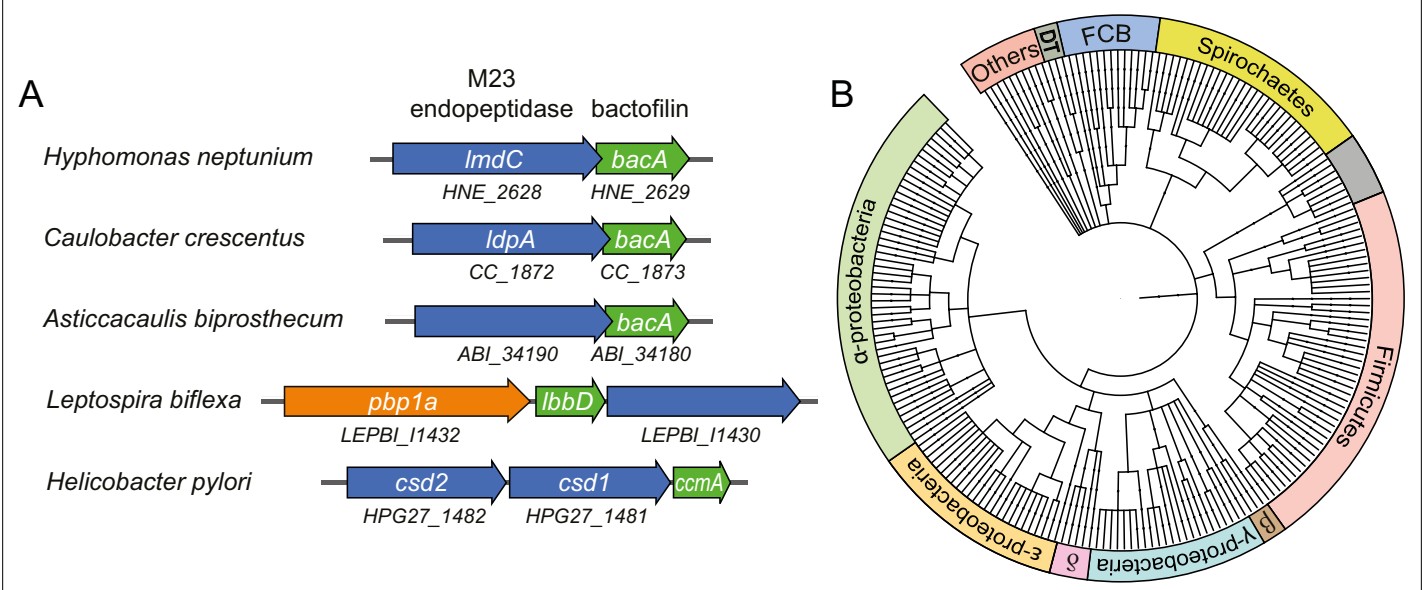

**Figure 7.** Putative operons of including adjacent BacA and M23 peptidase genes are widely conserved among bacteria. (**A**) Arrangement of *bacA* and *lmdC* genes in species whose BacA homolog has a proven role in cell morphogenesis. (**B**) Co-conservation of BacA and M23 peptidase genes across different bacterial phyla. Bioinformatic analysis was used to identify genomes that contained a bactofilin gene located next to an M23 peptidase gene. After retrieval of the taxonomy IDs of the corresponding species from the National Center for Biotechnology Information (NCBI) website, a phylogenetic tree of the species was created using the iTOL server (*Letunic and Bork, 2021*). Abbreviations: DT (Deinococcus-Thermus group), FCB (Fibrobacteres-Chlorobi-Bacteroidetes group), β (β-proteobacteria), δ (δ-proteobacteria). The full list of species used and details on the genes identified are provided in *Figure 7—source data 1*.

The online version of this article includes the following source data for figure 7:

**Source data 1.** List of bactofilin-M23 peptidase gene clusters identified in bacterial genomes.

M23 peptidase domain. This analysis identified a total of 226 species from a wide variety of bacterial phyla (*Figure 7B* and *Figure 7—source data 1*), suggesting a conserved functional association between bactofilins and M23 peptidases.

## The M23 peptidase LmdC of *H. neptunium* has peptidoglycan hydrolase activity in vitro

Like many of its homologs, *H. neptunium* LmdC is a predicted bitopic membrane protein with a short N-terminal cytoplasmic region, a transmembrane helix, large periplasmic region composed of a predicted coiled-coil domain, and a C-terminal M23 peptidase domain (*Figure 8A*). Members of the M23 peptidase family usually have Zn+-dependent hydrolase activity and cleave bonds within the peptide side chains of the peptidoglycan meshwork (*Firczuk et al., 2005*; *Grabowska et al., 2015*). However, there are also various representatives that have lost their enzymatic activity because of mutations in residues required for metal cofactor binding and have instead adopted regulatory roles in cell wall biosynthesis (*Figueroa-Cuilan et al., 2021*; *Goley et al., 2010*; *Gurnani Serrano et al., 2021*; *Möll et al., 2010*; *Poggio et al., 2010*; *Uehara et al., 2010*). An amino acid alignment showed that the M23 peptidase domain of LmdC features all residues reported to be critical for $Zn^{2+}$ binding, suggesting that it could act as a genuine peptidoglycan hydrolase (*Figure 8B*). In a structural model of LmdC generated with AlphaFold2 (*Jumper et al., 2021*), the periplasmic coiled-coil and M23 peptidase domains form an elongated, rigid unit with two flanking non-structured regions that is flexibly linked to the transmembrane helix (*Figure 8C*). Notably, LmdC homologs were shown to form a distinct clade among the M23 peptidases that is broadly conserved among species but particularly enriched in alpha- and deltaproteobacteria (*Figueroa-Cuilan et al., 2021*).

To clarify whether LmdC was indeed catalytically active, we purified a C-terminal fragment of the protein including the M23 peptidase domain, and assayed its activity in vitro. Peptidoglycan from *H. neptunium* shows only a low degree of cross-linkage and hardly any pentapeptides (*Cserti et al., 2017*),

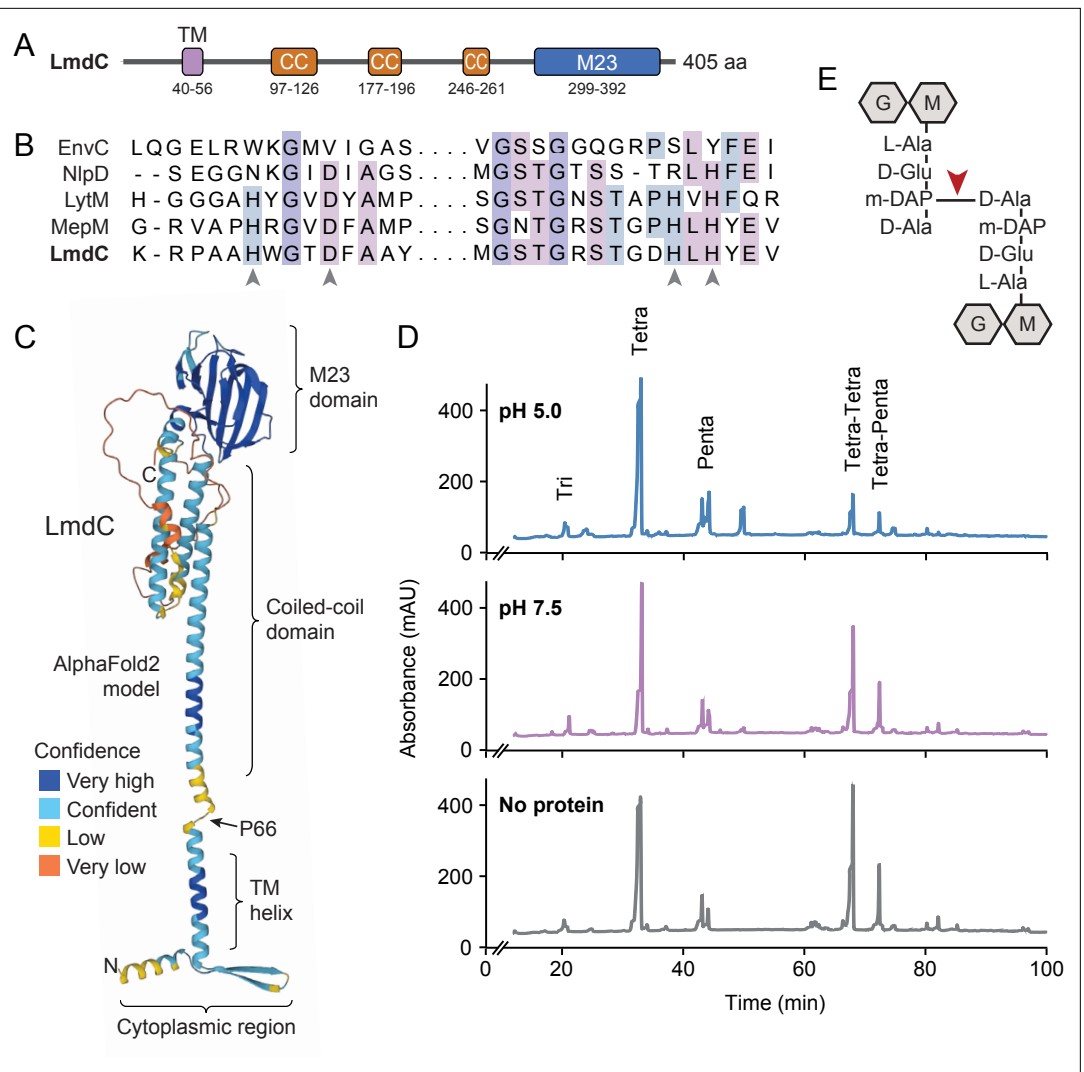

**Figure 8.** LmdC is a peptidoglycan hydrolase with DD-endopeptidase activity. (**A**) Predicted domain architecture of *H. neptunium* LmdC. The predicted positions of the transmembrane helix (TM), the three coiled-coil regions (CC), and the M23 peptidase domain are indicated. (**B**) Alignment of the amino acid sequences of multiple M23 peptidases showing the conservation of the catalytic residues in LmdC. Residues required to coordinate the catalytic $Zn^{2+}$ ion of LytM from *S. aureus* (**Firczuk et al., 2005**) are indicated by arrowheads. The proteins shown are EnvC from *E. coli* (UniProt: P37690), NlpD from *E. coli* (P0ADA3), LytM from *S. aureus* (O33599), MepM from *E. coli* (P0AFS9) and LmdC from *H. neptunium* (Q0BYX6). (**C**) Predicted molecular structure of *H. neptunium* LmdC, generated with Alphafold2 (**Jumper et al., 2021**). The different domains of the protein and the position of proline 66, which terminates the N-terminal fragment of LmdC used for the in vivo analyses in this study (LmdC$_{1-66}$), are indicated. (**D**) Representative HPLC traces showing the muropeptide profile of peptidoglycan treated with LmdC. Cell walls were incubated with the isolated M23 peptidase domain of LmdC at pH 5.0 or pH 7.5. Subsequently, muropeptides were released with cellosyl, reduced, and separated by HPLC. A sample lacking LmdC (No protein) was analyzed as a reference. The nature of the major products is indicated above the peaks. Tri, Tetra, and Penta stand for N-acetylglucosamine-N-acetylmuramitol tripeptide, tetrapeptide, and pentapeptide, respectively. (**E**) Structure of a Tetra-Tetra muropeptide. Abbreviations: G (N-acetylglucosamine), M (N-acetylmuramic acid). The cleavage site of LmdC is indicated by a red arrowhead.

likely due to a high level of autolytic activity. We, therefore, used normally crosslinked, pentapeptide-enriched peptidoglycan from *E. coli* strain D456 (lacking PBPs 4, 5, and 6) (**Edwards and Donachie, 1993**) as a substrate to enable a comprehensive analysis of the cleavage specificity of LmdC. Upon treatment with the protein, the relative proportion of dimeric muropeptide species (Tetra-Tetra and Tetra-Penta) strongly decreased, regardless of the length of the crosslinked peptides and without the

formation of major additional monomer peaks (*Figure 8D*). The activity of LmdC was particularly high at a pH value of 5, but the physiological significance of this effect remains to be determined. Collectively, these results demonstrate that LmdC is a DD-endopeptidase cleaving the bond between the meso-diaminopimelic acid residue of one peptide and D-alanine at position 4 of the other peptide, thereby reducing the degree of cross-linkage within the peptidoglycan layer (*Figure 8E*).

## LmdC is critical for morphogenesis in *H. neptunium* and physically interacts with BacA

To clarify whether LmdC was also required for proper growth in *H. neptunium*, we aimed to generate a mutant strain lacking LmdC activity. However, all attempts to delete the *lmdC* gene or the entire putative *lmdC-bacA* operon or to generate non-functional truncated *lmdC* alleles were unsuccessful, in line with a previous report suggesting that *lmdC* is essential for viability (*Cserti et al., 2017*). It was also not possible to generate a conditional *lmdC* mutant producing the gene under the control of a copper-inducible promoter, suggesting that its expression level needs to be precisely regulated.

To address this issue, we developed a CRISPR interference (CRISPRi) system (*Larson et al., 2013*) that enabled the inducible down-regulation of target genes in *H. neptunium*. It is based on a newly constructed integrative plasmid (pdCas9Entry) that carries the gene for a nuclease-deficient variant of *Streptococcus pyogenes* Cas9 (dCas9) under the control of a copper-inducible promoter as well as a cassette for the construction and constitutive expression of a gene-specific small guide RNA (sgRNA) (*Figure 9A* and *Figure 9—figure supplement 1A*). A derivative of this plasmid encoding an sgRNA directed against the 5' region of the *lmdC* gene was then inserted into the *H. neptunium* chromosome (*Figure 9B*). Induction with copper led to the formation of an sgRNA-dCas9 complex that targeted the non-coding strand of *lmdC*, generating a road-block that prevented the movement of RNA polymerase and, thus, gene expression (*Figure 9C and D*). Importantly, the block of *lmdC* expression led to cell shape defects very similar to those observed for the Δ*bacA* mutant, as reflected by a large proportion of distorted and amorphous cells, which still showed close-to-normal growth rates (*Figure 9E and F* and *Figure 9—figure supplement 1B and C*). Induced cells of a control strain carrying the empty pdCas9Entry plasmid did not show any obvious phenotype, indicating that the presence of dCas9 alone did not have any adverse effects on cell growth or viability (*Figure 9D and E* and *Figure 9—figure supplement 1C*). These results suggest that LmdC and BacA are part of the same morphogenetic pathway.

To further investigate whether the peptidoglycan hydrolase LmdC was functionally linked to bactofilins in *H. neptunium*, we aimed to conduct in vivo co-localization studies. However, all attempts to generate functional fluorescent protein fusions to full-length or C-terminally truncated LmdC failed, because the fusion proteins were either unstable or did not produce any fluorescence signal in vivo. As a complementary approach, we analyzed the interaction between LmdC and BacA in vitro by biolayer interferometry. To this end, a synthetic peptide comprising the predicted cytoplasmic region of LmdC (amino acids 1–38) was immobilized on a biosensor and probed with increasing concentrations of purified BacA protein. The results showed that BacA interacts with the LmdC peptide with an apparent equilibrium dissociation constant ($K_D$) of ~15 μM. By contrast, an unrelated peptide used as a negative control was not bound with appreciable affinity (*Figure 9G and H*). This result confirms a direct interaction between BacA and LmdC and identifies the N-terminal cytoplasmic region of LmdC as the interaction determinant for BacA association.

## *R. rubrum* BacA recruits LmdC to the inner curve of the cell to modulate cell curvature

Our bioinformatic analysis suggests that bactofilins and M23 peptidases may be functionally associated in a large number of species. To further explore this possibility, we turned our efforts to the spiral-shaped bacterium *Rhodospirillum rubrum*, which contains a putative *lmdC-bacA* operon similar to that in *H. neptunium* (*Munk et al., 2011*; *Figure 10A*). The *R. rubrum* BacA homolog Rru_A1867 (BacA$_{Rs}$) is 37% identical (57% similar) to BacA of *H. neptunium* and also consists of a central Bactofilin A/B domain flanked by terminal non-structured regions (*Figure 10B*). The LmdC homolog Rru_A1868 (LmdC$_{Rs}$) is 39% identical (55% similar) to *H. neptunium* LmdC and has a similar predicted molecular structure (*Varadi et al., 2022*). The deletion of *bacA$_{Rs}$* led to a noticeable increase in cell curvature (*Figure 10C* and *Figure 10—figure supplement 1A*), as also reflected in a significant increase in cell

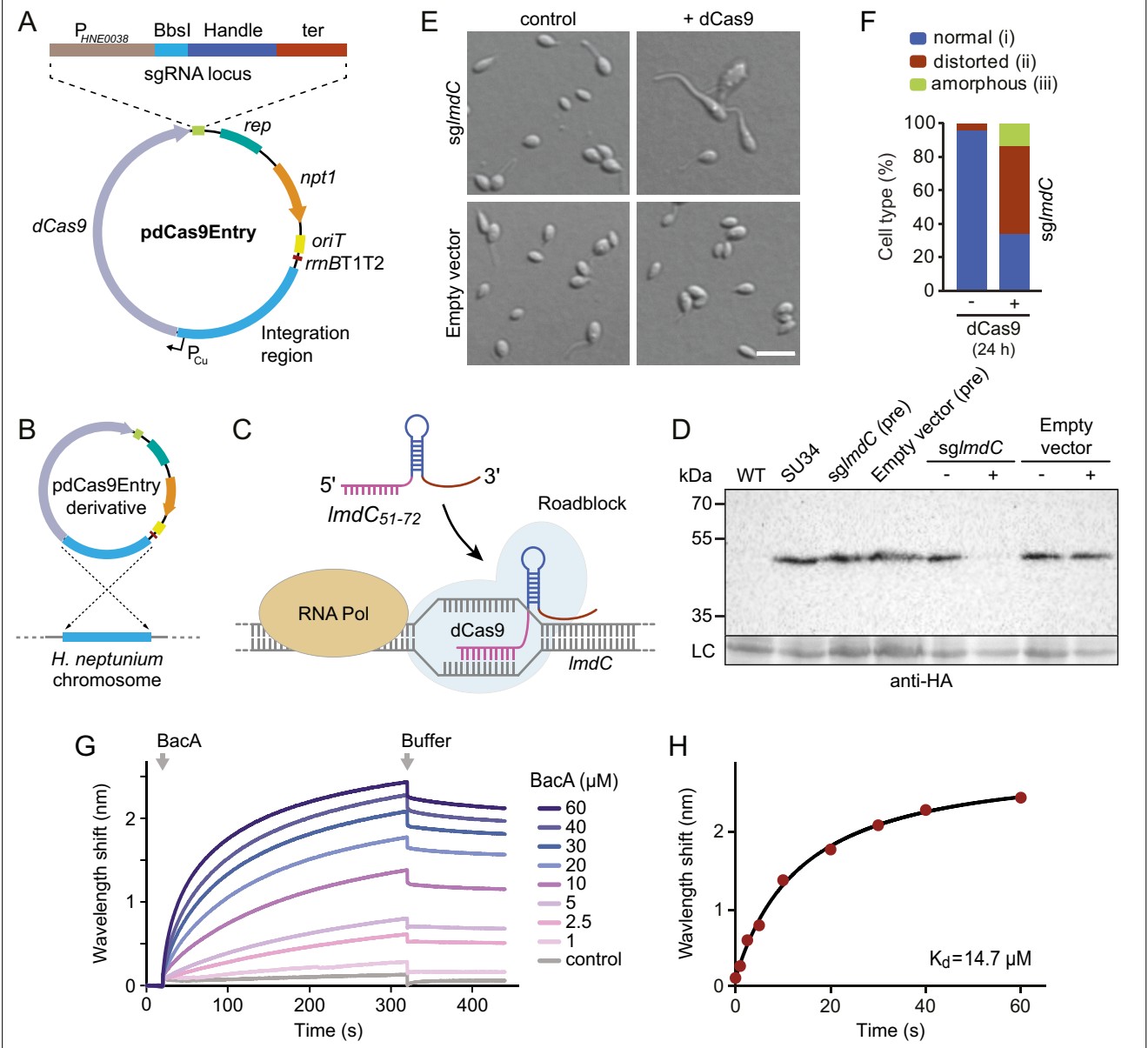

**Figure 9.** LmdC interacts with BacA to mediate *H. neptunium* morphogenesis. (**A**) Schematic representation of the integrative plasmid pdCas9Entry, enabling CRISPR interference (CRISPRi) in *H. neptunium*. The plasmid harbors a chromosomal fragment containing the copper-inducible $P_{Cu}$ promoter, followed by a gene encoding a nuclease-deficient variant of Cas9 (dCas9). It also contains an sgRNA expression cassette composed of the constitutive $P_{HNE0038}$ promoter, a BbsI cloning site for the insertion of target-specific DNA fragments, a sequence encoding an optimized small guide RNA (sgRNA) handle (**Chen et al., 2013**), and a transcriptional terminator (ter). Other features of the plasmid are a replication initiator gene (*rep*), a kanamycin resistance gene (*npt1*), an origin of transfer (*oriT*), and a strong transcriptional terminator (*rrnB*T1T2) that is placed upstream of the $P_{Cu}$-containing fragment. The DNA sequence of the sgRNA expression cassette is given in **Figure 9—figure supplement 1**. (**B**) Schematic showing the integration of a pdCas9Entry derivative into the *H. neptunium* chromosome by single homologous recombination at the $P_{Cu}$ locus. (**C**) Mechanism of the CRISPRi-mediated repression of *lmdC* expression. In the presence of inducer, dCas9 interacts with a constitutively produced sgRNA that targets bases 51–72 of the non-coding strand of *lmdC* (sg*lmdC*). The formation of the interference complex creates a roadblock that blocks the transcription of *lmdC* by RNA polymerase. (**D**) Immunoblot analysis verifying the CRISPRi-mediated depletion of LmdC. Strain SU34 (*lmdC::lmdC-HA*), which produces a fully functional variant of LmdC carrying the HA affinity tag inserted in between the transmembrane helix and the coiled-coil domain, was transformed with pdCas9Entry (empty vector) or plasmid pJH13 (sg*lmdC*). The resulting strains (SP249 and SP236) were grown for 24 hr in the absence (-) or presence (+) of 0.3 mM CuSO₄ and subjected to immunoblot analysis with an anti-HA antibody. The wild-type strain (LE670), the parental strain SU34 as well as the uninduced pre-cultures (pre) of strains SP249 and SP236 used for the experiment were analyzed as controls. A non-specific band served as a loading control (LC). (**E**) Morphology of *H. neptunium* cells depleted of LmdC. Shown are representative cells of strains SP236 (*lmdC::lmdC-HA* pJH13) (sg*lmdC*) and SP249 (*lmdC::lmdC-HA* pCas9Entry) (empty vector) grown for 24 hr in the absence (control) or presence (+dCas9) of 0.3 mM CuSO₄. Cells were

*Figure 9 continued on next page*

*Figure 9 continued*

imaged by DIC and fluorescence microscopy. Bar: 2 µm. (**F**) Quantification of the proportion of phenotypically abnormal stalked and budding cells in the cultures of strain SP236 (*lmdC::lmdC-HA* pJH13) analyzed in panel E (n=100 cells per condition). (**G**) Bio-layer interferometric analysis of the interaction between cytoplasmic domain of LmdC and BacA. Sensors derivatized with a biotinylated synthetic peptide comprising the N-terminal cytoplasmic region of LmdC (amino acids 1–38) were probed with the indicated concentrations of BacA. After the association of BacA, the sensors were transferred to protein-free buffer to induce BacA dissociation. The interaction kinetics were followed by monitoring the wavelength shifts resulting from changes in the optical thickness of the sensor surface during association or dissociation of the analyte. The extent of non-specific binding of BacA to the sensor surface was negligible (control). (**H**) Affinity of the BacA-LmdC interaction. The maximal wavelength shifts measured in the experiment shown in panel G were plotted against the corresponding BacA concentration. The data were fitted to a one-site binding model, yielding an apparent $K_D$ value of ~15 µM.

The online version of this article includes the following source data and figure supplement(s) for figure 9:

**Source data 1.** Raw images for the immunoblot analysis in *Figure 9D*.

**Figure supplement 1.** Characterization of the phenotype induced by the CRISPRi-mediated depletion of LmdC in *H. neptunium*.

sinuosity (*Figure 10D*). A very similar effect was observed for cells lacking *lmdC$_{Rs}$* or both *bacA$_{Rs}$* and *lmdC$_{Rs}$*, indicating that the two gene products act in the same pathway and jointly contribute to cell shape maintenance in *R. rubrum* (*Figure 10C and D* and *Figure 10—figure supplement 1A*). In the two mutant strains, the global composition of peptidoglycan was largely unchanged, supporting the notion that the BacA$_{Rs}$-LmdC$_{Rs}$ pathway modifies the cell wall only locally or acts by modulating the activity of the standard cell elongation machinery (*Figure 10—figure supplement 1B* and *Supplementary file 2*).

Complementation studies showed that the expression of a plasmid-borne copy of *bacA* under the control of a weak constitutive promoter attenuated the curvature defect, albeit only partially, likely due to inadequate expression levels (*Figure 10—figure supplement 2*). The generation of an analogous plasmid to complement the deletion of *lmdC$_{Rs}$* failed, because it was not possible to introduce the construct into *Escherichia coli* for plasmid propagation, suggesting that the expression of *lmdC$_{Rs}$* was toxic to the cells. To further test this hypothesis, we constructed a plasmid enabling the expression of the gene under a tightly controlled, inducible promoter in *E. coli*. Upon induction, a large part of the cell population lysed, whereas control cells carrying an empty plasmid did not show any phenotypic defects (*Figure 10—figure supplement 3*). These results indicate that, similar to its *H. neptunium* homolog, LmdC$_{Rs}$ is an active enzyme with peptidoglycan hydrolase activity.

To further test the hypothesis that LmdC was functionally linked to bactofilin in *R. rubrum*, we set out to perform in vivo co-localization studies. However, all attempts to generate fluorescently labeled variants of full-length LmdC failed, because the corresponding *R. rubrum* expression plasmids appeared to be toxic to *E. coli*. Importantly, our in vitro studies had shown that the interaction of LmdC with the bactofilin cytoskeleton was mediated by its N-terminal cytoplasmic region (*Figure 9G and H*). To avoid the issues caused by the full-length protein, we, therefore, generated a reporter fusion encoding only the predicted N-terminal cytoplasmic region and the transmembrane helix (amino acids 1–80) of LmdC$_{Rs}$ fused to the green fluorescent protein mNeonGreen (*Shaner et al., 2013*) (LmdC$^N_{Rs}$-mNG) (*Figure 10—figure supplement 4A–C*). A low-copy plasmid carrying the corresponding allele under the control of the native *lmdC$_{Rs}$* promoter was then introduced into an *R. rubrum* Δ*lmdC$_{Rs}$* mutant producing a fully functional fusion of BacA$_{Rs}$ to the red fluorescent protein mCherry (BacA$_{Rs}$-CHY) (*Shaner et al., 2004*) in place of the wild-type protein (*Figure 10—figure supplement 2*). We found that BacA$_{Rs}$-CHY and LmdC$^N_{Rs}$-mNG were consistently localized to the same subcellular positions, forming patchy or filamentous structures that were preferentially, but not exclusively, positioned at the inner curve of the cell (*Figure 10E*). A very similar behavior was observed in the presence of wild-type LmdC$_{Rs}$ (*Figure 10—figure supplement 4D*). To further validate the enrichment of the two fusion proteins at the inner curve, we quantified their fluorescence signals along the inner and outer curves of multiple individual cells, concatenated the resulting fluorescence profiles, and ordered them according to their total length. In both cases, cells consistently displayed higher signal intensities at the inner curve (*Figure 10F and G*), supporting the notion that BacA$_{Rs}$-LmdC$_{Rs}$ assemblies promote cell wall biosynthesis in the inner curve to reduce overall cell curvature. Notably, a BacA$_{Rs}$-mNG fusion showed a reduced enrichment in the absence of LmdC$_{Rs}$, whereas LmdC$^N_{Rs}$-mNG completely lost its asymmetric distribution in the Δ*bacA$_{Rs}$* background (*Figure 10H–J*). These results identify BacA$_{Rs}$ polymers as recruitment platforms for LmdC$_{Rs}$, although an interaction between the two proteins appears to be required for normal BacA$_{Rs}$ localization.

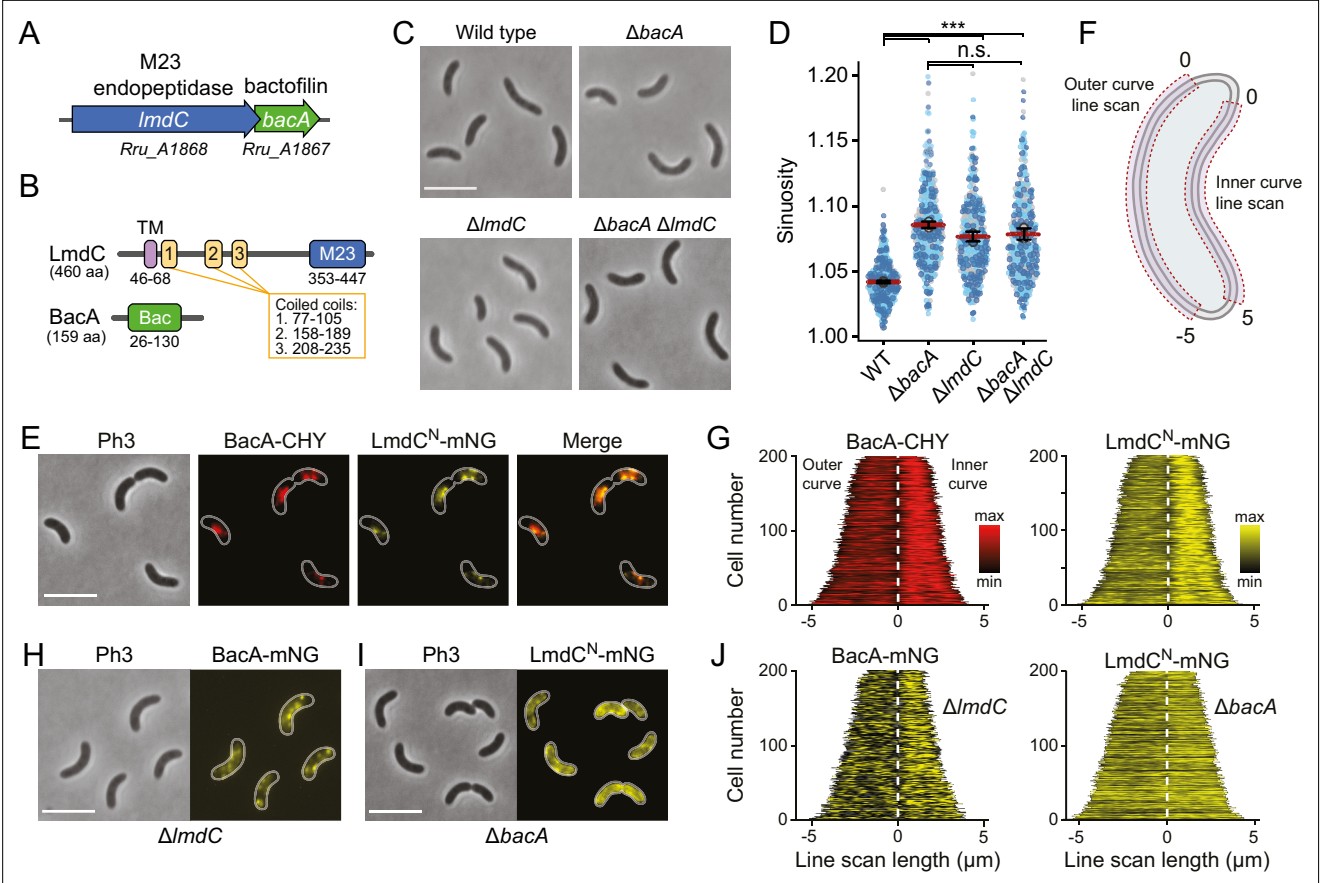

**Figure 10.** BacA and LmdC contribute to cell morphogenesis in *R. rubrum*. (**A**) Schematic representation of the *bacA-lmdC* operon in *R. rubrum*. (**B**) Domain organization of LmdC and BacA from *R. rubrum*. LmdC$_{Rs}$ consists of a transmembrane helix (TM) followed by a coiled-coil-rich region and a C-terminal M23 peptidase domain (**M23**). The bactofilin polymerization domain (green box) is flanked by non-structured N- and C-terminal regions. Numbers indicate the first and last amino acid of a domain. (**C**) Phenotypes of *R. rubrum* wild-type (**S1**), Δ*bacA*$_{Rs}$ (SP70), Δ*lmdC*$_{Rs}$ (SP68), and Δ*bacA*$_{Rs}$ Δ*lmdC*$_{Rs}$ (SP116) cells, imaged using phase-contrast microscopy. Bar: 5 μm. (**D**) Superplots showing the distribution of cell sinuosities in populations of the indicated *R. rubrum* strains. Small dots represent the data points, large dots represent the median values of the three independent experiments shown in the graphs (dark blue, light blue, gray). The mean of the three median values is indicated by a horizontal red line. Whiskers represent the standard deviation. ***p<0.005; ns, not significant; unpaired two-sided Welch's t-test. (**E**) Co-localization of BacA$_{Rs}$-CHY and LmdC$^N$$_{Rs}$-mNG in the Δ*lmdC*$_{Rs}$ background (SP119). The Pearson's Correlation Coefficient (PCC) of the two fluorescence signals in a random subpopulation of cells is 0.916. Scale bar: 5 μm. (**F**) Schematic showing the approach used to quantify the distribution of proteins along the outer and inner curve of a cell in panels G and J. (**G**) Demographs showing the enrichment of BacA-CYH and LmdC$^N$-mNG at the inner curve the populations of strain SP119 analyzed in panel E (n=200 cells per strain). The white dashed line represents the point of transition from the outer to the inner curve. (**H**) Localization of BacA$_{Rs}$-mNG in the Δ*lmdC*$_{Rs}$ background (SP98). Bar: 5 μm. (**I**) Localization of LmdC$_{Rs}$-mNG in the Δ*bacA*$_{Rs}$ background (SP118). Bar: 5 μm. (**J**) Demographs showing the outer-versus-inner curve distribution of BacA$_{Rs}$-mNG in the Δ*lmdC*$_{Rs}$ background (SP98; n=200 cells) and LmdC$^N$$_{Rs}$-mNG in the Δ*bacA*$_{Rs}$ background (SP118; n=200 cells). The white dashed line represents the point of transition from the outer to the inner curve.

The online version of this article includes the following source data and figure supplement(s) for figure 10:

**Figure supplement 1.** Analysis of the role of BacA and LmdC in *R. rubrum* morphogenesis.

**Figure supplement 2.** Cell lengths of *R. rubrum* strains.

**Figure supplement 3.** Lysis of *E. coli* upon heterologous expression of full-length *R. rubrum lmdC*.

**Figure supplement 4.** Analysis of *R. rubrum* strains producing fluorescent protein fusions.

**Figure supplement 4—source data 1.** Raw images for the immunoblot analyses in *Figure 10—figure supplement 4B and C*.

## LmdC recruitment depends on the conserved cytoplasmic β-hairpin motif

The N-terminal cytoplasmic region of LmdC$_{Rs}$ contains a β-hairpin motif (*Figure 11A*) that is conserved among LmdC homologs. To determine whether this structural feature was involved in BacA binding,

we exchanged two small clusters of amino acids that protruded from its surface and might thus serve as interaction determinants, using LmdC$^N_{Rs}$-mNG as a reporter. Cluster I (R25, H26, L27) was located within the first β-strand, whereas cluster II (R30, S31) was close to the β-turn (*Figure 11B*). The modifications in cluster I completely abolished the enrichment of LmdC at the inner curve of the cell, indicating that the β-hairpin motif is a critical part of the BacA binding site (*Figure 11C and D*). Notably, the loss of interaction between the two proteins also led to a reduced inner-curve enrichment of BacA$_{Rs}$-CHY, as previously observed in the Δ*lmdC$_{Rs}$* background (*Figure 10I and J*). This observation confirms that LmdC$_{Rs}$ binding is required for BacA to adopt its proper spatial organization. Similar defects were caused by the modification of cluster II (*Figure 11C and E*). However, in this case, BacA$_{Rs}$-CHY was also no longer enriched at the inner curve, suggesting that the mutant protein may still show a residual, yet aberrant interaction with BacA$_{Rs}$ that has a negative effect on BacA$_{Rs}$ localization. Collectively, these findings confirm a functional association between the bactofilin cytoskeleton and LmdC$_{Rs}$ in *R. rubrum*. Moreover, they support the notion that bactofilins and LmdC-like M23 peptidases form a conserved morphogenetic module that is involved in the establishment of complex bacterial cell shapes.

## Discussion

Although bactofilins are widely conserved among bacteria, their cellular functions are still incompletely understood. Here, we show that BacA homologs critically contribute to cell shape determination in

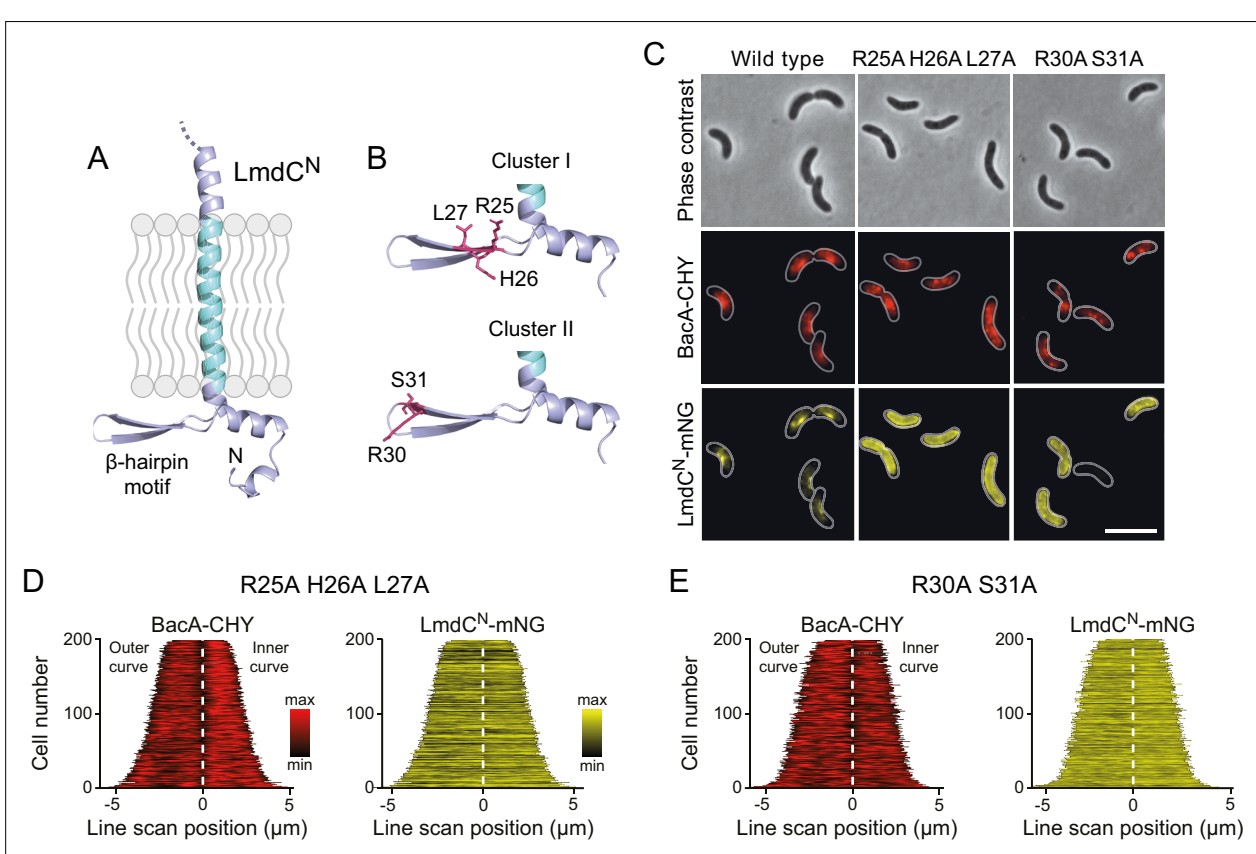

**Figure 11.** The β-hairpin in the cytoplasmic region of LmdC is critical for BacA binding. (**A**) Predicted structure of LmdC$^N$, comprising the first 80 amino acids of LmdC. The transmembrane domain is colored light blue. The cytoplasmic domain includes a conserved β-hairpin motif. (**B**) Clusters of amino acids mutated for the analysis of BacA-LmdC interaction. Mutated amino acids are shown as red sticks. (**C**) Localization of BacA$_{Rs}$-CHY and the indicated LmdC$^N_{Rs}$-mNG variants in the Δ*lmdC$_{Rs}$* background. The Pearson's Correlation Coefficient (PCC) of the two fluorescence signals in a random subpopulation of cells is 0.914 for the wild-type fusion protein (SP119), 0.887 for the R25A-H26A-L27A variant (SP237) and 0.827 for the R30A-S31A variant (SP238). Scale bar: 5 μm. (**D,E**) Demographs showing the distribution of BacA-mCherry and (**D**) R25A-H26A-L27A variant (SP237) or R31A-S31A variant (SP238) of LmdC$^N$-mNG along the outer and inner curve of the cell (n=200 cells per strain). The white line marks the point of transition from the outer and the inner curve. The analysis was performed as described in *Figure 10F*.

two morphologically distinct members of the alphaproteobacteria, the stalked budding bacterium *H. neptunium* and the spiral-shaped bacterium *R. rubrum*. In both cases, they functionally interact with LmdC-type DD-endopeptidases to promote local changes in the pattern of peptidoglycan biosynthesis. Bactofilins thus complement the activities of the MreB and FtsZ cytoskeletons in the regulation of cell growth and act as versatile cell shape modifiers that facilitate the establishment of complex bacterial morphologies.

In *H. neptunium*, the bactofilin cytoskeleton localizes dynamically to the stalk base and the bud neck, promoting stalk growth and subsequent bud formation (*Figure 4*). While flanked by zones of active growth, the stalk itself is usually devoid of cell wall biosynthetic machinery (*Cserti et al., 2017*; *Figure 5*). In bactofilin-deficient cells, however, the stalk and bud growth zones are no longer confined and expand into the stalk envelope, leading to stalk widening and the formation of irregular bulges due to uncontrolled peptidoglycan incorporation and bud expansion (*Figure 2* and *Figure 5*). Bulges or larger amorphous segments at some point separate from the mother cell and then continue to grow as independent cells, indicating that chromosome replication and segregation as well as cell division still occur under these conditions. Together, these observations suggests that bactofilin polymers establish barriers that normally prevent the entry of elongasome complexes from the mother cell or nascent bud compartments into the stalk.

The precise mechanism underlying the function of bactofilin in cell morphogenesis remains to be clarified. It is conceivable that bactofilin polymers act by tethering peptidoglycan biosynthetic proteins or establishing physical barriers that hinder the mobility of elongasome complexes (*Figure 12A*). However, we observed a close functional association of bactofilins with M23 peptidases that appears to be widely conserved among species, suggesting that their activity may be intimately tied to cell wall hydrolysis. The transition zones between the stalk and the adjacent mother cell and bud compartments are characterized by a high degree of positive inner cell curvature, which is in stark contrast to the negative curvature in the remaining parts of the cell. The hydrolytic activity of *H. neptunium* LmdC may be critical to remodel the cell wall in these zones and thus facilitate the extrusion of the stalk. Importantly, MreB is thought to move along regions of negative inner cell curvature (*Hussain et al., 2018*; *Wong et al., 2019*). The positively curved transition zones generated by the bactofilin-LmdC assemblies could, therefore, represent topological barriers that are difficult to cross by elongasome complexes, thereby helping to restrict their movement to the mother cell and bud compartments. Moreover, the narrow diameter stalks and their exceedingly high degree of negative inner cell curvature may reduce the stability of elongasome complexes that may form spontaneously in the stalk region. These effects may also be important during the stalk elongation phase following cell division, when bactofilin assemblies are no longer present at the stalk base.

Notably, a role in stalk formation has also been reported for the bactofilin homologs of *C. crescentus* and *A. biprosthecum*. In *C. crescentus*, the major bactofilin BacA was shown to act as a localization determinant for the bifunctional penicillin-binding protein PbpC, a cell wall synthase that contributes to stalk elongation (*Kühn et al., 2010*) and the proper sorting of a stalk-specific protein (*Hughes et al., 2013*). However, the absence of BacA only leads to a moderate reduction in stalk length and leaves overall cell shape unaffected, indicating that this protein has only a minor role in *C. crescentus* stalk formation. In *A. biprosthecum*, by contrast, the deletion of *bacA* completely abolishes stalk formation in rich medium and leads to the formation of wide cellular protrusions, named pseudostalks, under phosphate-limiting conditions (*Caccamo et al., 2020*). Similar to the amorphous extensions observed for the *H. neptunium* Δ*bacAD* mutant, these structures grow through disperse incorporation of new peptidoglycan and develop into viable offspring. Their formation was also attributed to the entry of cell wall biosynthetic proteins into nascent stalks (*Caccamo et al., 2020*), suggesting that the *H. neptunium* and *A. biprosthecum* BacA homologs share the same barrier function during stalk formation. However, in *H. neptunium*, another layer of regulation has been added in which bactofilins establish a second barrier close to the stalk tip that enables the formation of stalk-terminal buds. Bud expansion requires the relocation of elongasome complexes from the mother cell to the nascent bud at the onset of bud formation, as likely reflected by the fact that components of the elongasome (MreB, RodZ) and new cell wall biosynthesis can be detected within the stalk during a short time window at the end of the stalk elongation phase (*Cserti et al., 2017*). This process may be facilitated by the (partial) disassembly of the bactofilin complex at the stalk base and the diffusion of its components to the bud compartment, but the mechanistic details remain to be investigated. Another open

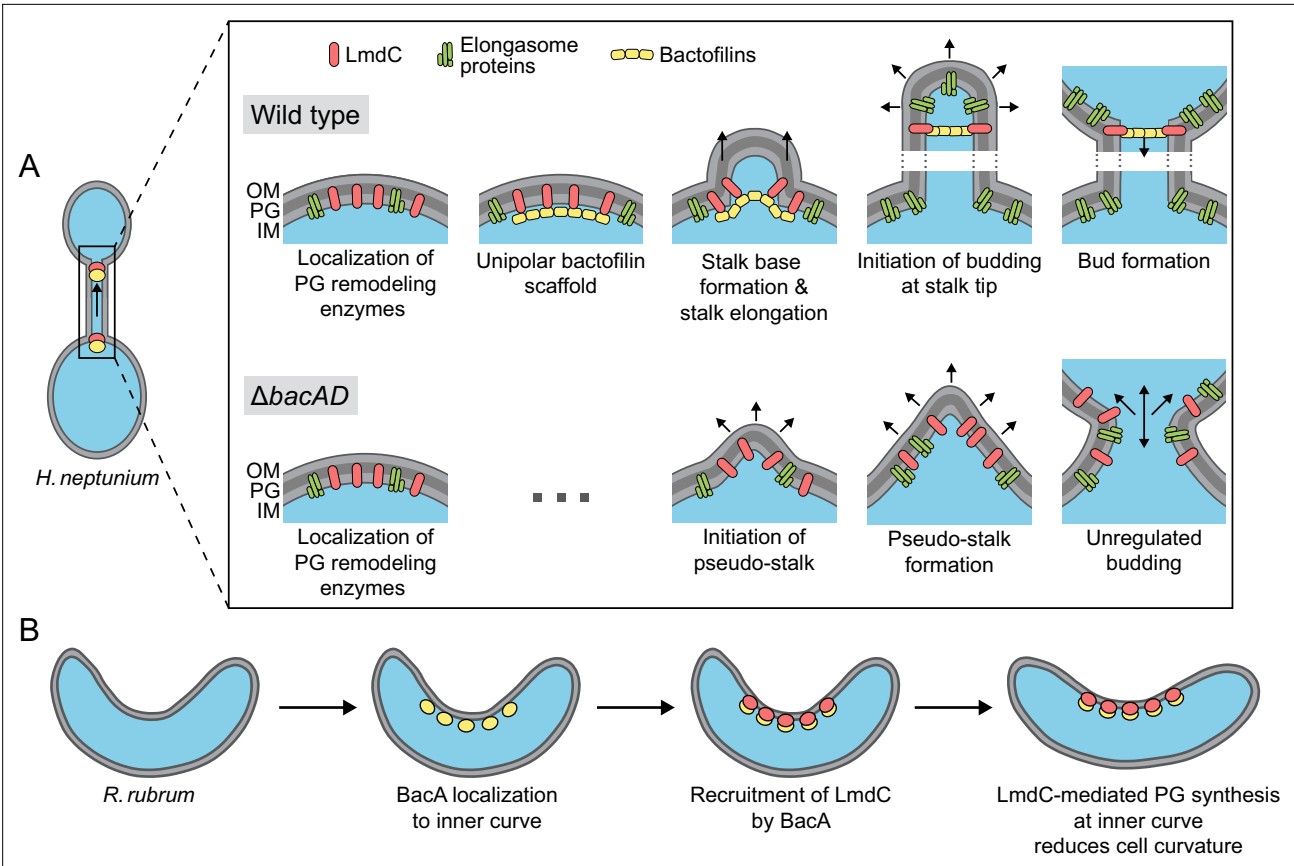

**Figure 12.** Model of the roles of BacA and LmdC in *H. neptunium* and *R. rubrum* cell morphogenesis. (**A**) Cell morphogenesis in *H. neptunium* wild-type (top) and Δ*bacAD* cells (bottom). In wild-type cells, BacA polymers form a complex with the DD-endopeptidase LmdC at the future stalked pole. The hydrolytic activity of LmdC, possibly stimulated by its interaction with BacA, helps to curve the cell wall at the incipient stalk base and thus determine stalk morphology. The zone of high positive inner cell curvature and, potentially, the physical barrier constituted by the bactofilin polymer prevent the movement of elongasome complexes from the mother cell body into the stalk, thereby limiting peptidoglycan biosynthesis to the stalked cell pole. At a later stage, the bactofilin-LmdC complex localizes close to the tip of the stalk and again generates a ring-shaped zone of positive cell curvature, thereby promoting the remodeling of the stalk tip into a spherical bud. At the onset of the budding process, elongasome complexes accumulate in the nascent bud by a so-far unknown mechanism. The positively curved bud neck and the bactofilin polymers prevent the movement of elongasome complexes from the bud into the stalk, thereby limiting cell growth to the bud compartment. In the Δ*bacAD* mutant, cells fail to concentrate LmdC at the future stalk base and bud neck. As a consequence, peptidoglycan biosynthesis is no longer limited to the different growth zones, leading to pseudo-stalk formation and unregulated bud expansion. (**B**) Bactofilin-mediated modulation of cell curvature in *R. rubrum*. BacA_Rs (yellow) recruits LmdC_Rs (red) to the inner curve of the cell. The hydrolytic activity of LmdC ultimately stimulates peptidoglycan biosynthesis at this position, leading to a reduction in cell helicity.

question concerns the factors that control the dynamic localization of bactofilin to the stalked pole and the future bud neck in the terminal segment of the stalk. In *A. biprosthecum*, the recruitment of BacA assemblies to the sites of stalk biosynthesis is dependent on the cell wall hydrolase SpmX (*Caccamo et al., 2020*). However, an *H. neptunium* mutant lacking this protein shows normal morphology (*Leicht et al., 2020*), suggesting the existence of a different localization determinant.

Intriguingly, bactofilins not only mediate stalk formation and stalk-terminal budding but also the formation of curved cell shapes. We show that both BacA and LmdC are required to establish the normal degree of cell curvature in *R. rubrum*, with their absence leading to hyper-curved cells (*Figure 10C and D*). This result suggests that the two proteins form an accessory module that counteracts the activity of a so-far unknown system responsible for generating spiral cell shape in *R. rubrum* (*Figure 12B*). Cell curvature was shown to promote cell motility (*Martínez et al., 2016*). It is tempting to speculate that the expression of *bacA* and *lmdC* could be controlled in response to external cues to ensure optimal cell curvature and, thus, fitness in different environments. Our results show that *R. rubrum* cells contain multiple BacA-LmdC complexes that are distributed along the entire cell

envelope but localize preferentially to the inner curve (*Figure 10E and G*). In this system, it is immediately obvious that BacA acts as a localization determinant for LmdC, since the preferential localization of LmdC to the inner curve is abolished in its absence (*Figure 10I and J*). To reduce cell curvature, the hydrolytic activity of LmdC must ultimately stimulate the insertion of new peptidoglycan at the inner curve of the cell, which increases the rate of cell elongation in this region and thus straightens the cell body. A similar mechanism may also be at work in the spirochete *L. biflexa*, where the loss of the bactofilin homolog LbbD, whose gene lies adjacent to a gene for putative M23 peptidase (LEPBI_I1430) (*Figure 7A*), was found to yield severely hyper-curved and more tightly twisted cells (*Jackson et al., 2018*). A different variation of this theme is found in *H. pylori*. There, the bactofilin homolog CcmA also forms multiple assemblies along the cell envelope, which interact with a membrane-spanning protein complex including an M23 peptidase (Csd1) homologous to LmdC (*Sichel et al., 2022*; *Taylor et al., 2020*). However, these assemblies are enriched at the outer curve of the cell, where they stimulate peptidoglycan synthesis to locally increase the rate of cell elongation over that at the inner curve, leading to twisting of the cell body.

Notably, in all systems characterized so far in molecular detail, the function of bactofilin-M23 peptidase complexes involves the stimulation of cell wall biosynthesis in a confined region of the cell envelope that entails a local change in cell envelope curvature. In *H. neptunium*, these complexes are only localized to narrow bands at the stalk base and bud neck, generating sharp bends at the transition zones between the stalk and the adjacent mother cell and bud compartments. In spiral-shaped bacteria, by contrast, they are scattered along the entire length of the cell, thereby establishing an elongated zone of increased longitudinal growth that changes overall cell curvature. The same mechanistic principle may also explain the morphogenetic role of bactofilins in so-far uninvestigated systems.

Collectively, our results underscore the role of bactofilins as versatile modulators of bacterial cell shape. In association with M23 peptidases, they form cell wall biosynthetic complexes that introduce local changes in cell envelope curvature. In doing so, they complement the activities of the tubulin and actin cytoskeletons in cell shape determination and thus expand the morphogenetic potential of bacteria, enabling the generation of complex cell shapes that go beyond the generic rod-like or spherical morphologies. It will be interesting to investigate the molecular function of bactofilins in a range of morphologically diverse species to obtain a comprehensive picture of the functionalities provided by these cytoskeletal proteins and the conservation of their mode of action.

## Materials and methods
### Media and growth conditions
*H. neptunium* LE670 (ATCC 15444) (*Leifson, 1964*) and its derivatives were grown in Artificial Sea Salt (ASS) medium at 28 °C under aerobic conditions, shaking at 210 rpm in baffled flasks. ASS medium was composed of 0.5% Bacto Peptone (Thermo Fisher Scientific, USA), 0.1% yeast extract, 1 mM $MgSO_4$, 0.5 mM $CaCl_2$ and 1.5% Instant Ocean Sea Salt (Spectrum Brands, USA), dissolved in deionized water. For solid media, 1.5% agar was added prior to autoclaving. When appropriate, antibiotics were used at the following concentrations (µg/mL in liquid/solid medium): rifampicin (1/2), kanamycin (100/200). The expression of genes under the control of the copper-inducible $P_{Cu}$ promoter or the zinc-inducible $P_{Zn}$ promoter (*Jung et al., 2015*) was induced by the addition of $CuSO_4$ or $ZnSO_4$ to the concentrations indicated in the text. To assess the growth of *H. neptunium*, cells were grown to exponential phase, diluted in fresh medium to an optical density at 580 nm ($OD_{580}$) of 0.02, and transferred into 24-well polystyrene microtiter plates (Becton Dickinson Labware, USA). Growth was then followed at 32 °C under double-orbital shaking in an EPOCH 2 microplate reader (BioTek, USA) by measuring the $OD_{580}$ at 20 min intervals.

*R. rubrum* S1 (ATCC 11170) (*Molisch, 1907*; *Pfennig and Truper, 1971*; *van Niel, 1944*) and its derivatives were grown in Bacto Tryptic Soy Broth (BD Diagnostic Systems, USA) at 28 °C under aerobic conditions, shaking at 210 rpm in Erlenmeyer flasks. When appropriate, media were supplemented with antibiotics at the following concentrations (µg/mL in liquid/solid medium): kanamycin (30/30), cefalexin (15/-).

*E. coli* strains were cultivated aerobically (shaking at 210 rpm) at 37 °C in LB medium. For plasmid-bearing strains, antibiotics were added at the following concentrations (µg/mL in liquid/solid medium):

kanamycin (30/50), rifampicin (25/50), ampicillin (50/200). To grow *E. coli* WM3064, media were supplemented with 2,6-diaminopimelic acid (DAP) to a final concentration of 300 µM.

## Determination of growth rates

Exponentially growing cultures were diluted to an $OD_{580}$ of ~0.01, transferred to a 96-well plate, and incubated at 30 °C with double-orbital shaking at 355 cpm in an Epoch 2 microplate reader (BioTek, Germany). Growth was monitored by measuring the $OD_{600}$ at 10 min intervals until the stationary phase was reached. $OD_{600}$ measurements obtained in the exponential phase were used to calculate the growth rate $\mu$ (increase in $OD_{600}$ per hour). The doubling time $t_d$ was then calculated using the equation $t_d = \ln(2)/\mu$.

## Plasmid and strain construction

The bacterial strains, plasmids, and oligonucleotides used in this study are listed in *Supplementary file 3*. *E. coli* TOP10 (Thermo Fisher Scientific, USA) was used as a host for cloning purposes. All plasmids were verified by DNA sequencing. *H. neptunium* and *R. rubrum* were transformed by conjugation using the DAP-auxotrophic strain *E. coli* WM3064 as a donor (*Jung et al., 2015*). The integration of non-replicating plasmids at the chromosomal $P_{Cu}$ or $P_{Zn}$ locus of *H. neptunium* was achieved by single homologous recombination (*Jung et al., 2015*). Gene replacement in *H. neptunium* and *R. rubrum* was achieved by double-homologous recombination using the counter-selectable *sacB* marker (*Cserti et al., 2017*). Proper chromosomal integration or gene replacement was verified by colony PCR.

## Live-cell imaging

Cells of *H. neptunium* and *R. rubrum* were grown to exponential phase in the appropriate medium and, if suitable, induced with $CuSO_4$ prior to imaging. For depletion experiments, cells were grown in the presence of inducer, washed three times with inducer-free medium, and then further cultivated in the absence of inducer for the indicated period of time prior to analysis. To acquire still images, cells were transferred onto 1% agarose pads (in water). For time-lapse analysis, cells were immobilized on pads made of 1% agarose in ASS medium, and the cover slides were sealed with VLAP (1:1:1 mixture of vaseline, lanolin, and paraffin) to prevent dehydration. Imaging was performed using a Zeiss Axio Imager.M1 microscope equipped with a Zeiss alpha Plan-Apochromat 100 x/1.40 Oil DIC objective and a pco.edge 3.1 sCMOS camera (PCO, Germany) or a Zeiss Axio Imager.Z1 microscope equipped with a Zeiss alpha Plan-Apochromat 100 x/1.46 Oil DIC M27 or a Plan-Apochromat 100 x/1.40 Oil Ph3 M27 objective and a pco.edge 4.2 sCMOS camera (PCO, Germany). An X-Cite 120PC metal halide light source (EXFO, Canada) and ET-DAPI, ET-YFP, or ET-TexasRed filter cubes (Chroma, USA) were used for fluorescence detection. Microfluidic experiments were performed using a CellASIC ONIX EV262 Microfluidic System, equipped with an F84 manifold and B04A microfluidic plates (Merck Millipore, Germany), which were flushed with PBS buffer for 30 min before usage. Exponentially growing cells were flushed into the flow cells, cultivated under continuous medium flow (0.4 ml/h), and imaged at regular intervals using the Axio Image.Z1 microscope described above. To visualize sites of ongoing peptidoglycan synthesis, cells were grown to the exponential phase, incubated for 1 min with 1 mM HADA, and washed four times in ASS medium prior to imaging.

Images were recorded with VisiView 3.3.0.6 (Visitron Systems, Germany) and processed with ImageJ (*Schneider et al., 2012*) and Adobe Illustrator CS6 (Adobe Systems, USA). The subcellular distribution of fluorescence signals was analyzed with BacStalk (*Hartmann et al., 2020*). Pearson's correlation coefficients were determined using the JACoP plug-in (*Bolte and Cordelières, 2006*) in ImageJ. Cell sinuosity was determined using the ImageJ plug-in MicrobeJ (*Ducret et al., 2016*), with each analysis performed in triplicate and 100 cells analyzed per strain and experiment. Results were displayed as SuperPlots (*Lord et al., 2020*) generated using the SuperPlotsOfData web application (*Goedhart, 2021*).

To analyze the distribution of fluorescently tagged BacA and LmdC variants along the outline of *R. rubrum* cells, the fluorescence intensity along the outer and inner curve was determined for 200 cells per strain using Fiji (*Schindelin et al., 2012*). After concatenation of the two fluorescence profiles obtained for each cell, the combined profiles were plotted as demographs, using R 4.2.3 (*R Development Core Team, 2012*) and the Cell-Profiles package (https://github.com/ta-cameron/Cell-Profiles/wiki/Vignette; *Cameron et al., 2014*).

## Single-particle tracking and diffusion analysis

Cells were cultivated overnight in ASS medium, transferred into fresh medium, and grown to exponential phase prior to imaging by slimfield microscopy (*Plank et al., 2009*). In this approach, the back aperture of the objective is underfilled by illumination with a collimated laser beam of reduced width, generating an area of ~10 µm in diameter with a light intensity high enough to enable the visualization of single fluorescent protein molecules at very high acquisition rates. The single-molecule level was reached by bleaching most molecules in the cell for 100–1,000 frames, followed by tracking of the remaining and newly synthesized molecules for ~3,000 frames. Images were taken at 30 ms intervals using an Olympus IX-71 microscope equipped with a UAPON 100 x/ NA 1.49 TIRF objective, a back-illuminated electron-multiplying charge-coupled device (EMCCD) iXon Ultra camera (Andor Solis, USA) in stream acquisition mode, and a LuxX 457–100 (457 nm, 100 mW) light-emitting diode laser (Omicron-Laserage Laserprodukte GmbH, Germany) as an excitation light source. The laser beam was focused onto the back focal plane and operated during image acquisition with up to 2 mW (60 W/cm$^2$ at the image plane). Andor Solis 4.21 software was used for camera control and stream acquisition. Prior to analysis, frames recorded before reaching the single-molecule level were removed from the streams, using photobleaching curves as a reference. Subsequently, the streams were cropped to an equal length of 2000 frames and the proper pixel size (100 nm) and time increment were set in the imaging metadata using Fiji (*Schindelin et al., 2012*). Single particles were tracked with u-track 2.2 (*Jaqaman et al., 2008*). Trajectories were only considered for further statistical analysis if they had a length of at least five steps. Data analysis was performed using SMTracker 2.0 (*Oviedo-Bocanegra et al., 2021*). An estimate of the diffusion coefficient and insight into the kind of diffusive motion exhibited were obtained from mean-squared-displacement (MSD)-versus-time-lag curves. In addition, the frame-to-frame displacements of all molecules in x and the y direction were fitted to a two-population Gaussian mixture model to determine the proportions of mobile and static molecules in each condition (*Oviedo-Bocanegra et al., 2021*).

## Transmission electron microscopy

To image *H. neptunium* cells, 10 µl of early exponential cell cultures were applied onto glow-discharged electron microscopy grids (Formvar/Carbon Film on 300 Mesh Copper; Plano GmbH, Germany) and incubated for 1 min at room temperature. The grid was manually blotted with Whatman filter paper to remove excess liquid. Subsequently, the cells were negatively stained for 5 sec with 5 µl of 1% uranyl acetate. After three washes with H$_2$O, the grids were dried and imaged in a 100 kV JEM-1400 Plus transmission electron microscope (JEOL, USA). To image BacA polymers, purified BacA-His6 was dialyzed against low-salt buffer (50 mM HEPES pH 7.2, 10 mM NaCl, 5 mM MgCl2, 0.1 mM EDTA, 1 mM ß-mercaptoethanol) for 16 hr. The protein was spotted onto carbon-coated grids and allowed to settle for 2 min. The grids were blotted dry, stained with 1:2 diluted supernatant of saturated 2% uranyl acetate (in H$_2$O) for 1 min, dried and imaged using a Zeiss CEM902 electron microscope, operated at 80 kV and equipped with a 2048 × 2048 pixel CCD camera. Image processing was carried out using Photoshop CS2 and Illustrator CS5 (Adobe Systems, USA).

## Immunoblot analysis

Antibodies against *R. rubrum* BacA were raised by immunization of rabbits with purified BacA-His$_6$ protein (Eurogentec, Belgium). Cells were harvested in the exponential growth phase. Immunoblot analysis was conducted as described previously (*Thanbichler and Shapiro, 2006*), using anti-BacA antiserum (1:10.000), a monoclonal anti-mNeonGreen antibody (Chromotek, Germany; Cat. #: 32f6; RRID: AB_2827566), a polyclonal anti-GFP antibody (Sigma, Germany; Cat. #: G1544; RRID: AB_439690), a polyclonal anti-mCherry antibody (BioVision, USA; Cat. #: 5993; RRID: AB_1975001) or a monoclonal anti-HA antibody (Merck Millipore, USA; Cat. #: 05–904; RRID: AB_11213751) at dilutions of 1:10,000, 1:1000, 1:10,000, 1:10,000 or 1:1000, respectively. Goat anti-rabbit immunoglobulin G conjugated with horse-radish peroxidase (Perkin Elmer, USA) or goat anti-mouse immunoglobulin G conjugated with horse-radish peroxidase (Sigma, Germany) were used as secondary antibodies. Immunocomplexes were detected with the Western Lightning Plus-ECL chemiluminescence reagent (Perkin Elmer, USA). The signals were recorded with a ChemiDoc MP imaging system (BioRad, Germany) and analyzed using Image Lab software (BioRad, Germany).

## Peptidoglycan analysis

Cultures of exponentially growing cells of the *H. neptunium* wild-type and its mutant derivatives EC28 (Δ*bacA*), EC23 (Δ*bacD*), and EC33 (Δ*bacAD*) were rapidly cooled to 4 °C and harvested by centrifugation at 16,000 × g for 30 min. The cells were resuspended in 6 ml of ice-cold $H_2O$ and added dropwise to 6 ml of a boiling solution of 8% sodium dodecylsulfate (SDS) that was stirred vigorously. After 30 min of boiling, the suspension was cooled to room temperature. Peptidoglycan was isolated from the cell lysates as described previously (*Glauner, 1988*) and digested with the muramidase cellosyl (kindly provided by Hoechst, Frankfurt, Germany). The resulting muropeptides were reduced with sodium borohydride and separated by HPLC following an established protocol (*Bui et al., 2009*; *Glauner, 1988*). The identity of eluted fragments was assigned based on the known retention times of muropeptides, as reported previously (*Cserti et al., 2017*; *Glauner, 1988*).

## Protein purification

To purify BacA-His$_6$, *E. coli* Rosetta(DE3)pLysS (Invitrogen) was transformed with pEC86 and grown in LB medium at 37 °C to an OD$_{600}$ of 0.8. Isopropyl-β-D-1-thiogalactopyranoside (IPTG) was added to a final concentration of 0.5 mM and the incubation was continued for another 3 hr. The cells were harvested by centrifugation for 15 min at 7500 × g and 4 °C and washed with buffer B2 (50 mM NaH$_2$PO$_4$, 300 mM NaCl, 10 mM imidazole, adjusted to pH 8.0 with NaOH). Subsequently, they were resuspended in buffer B2 containing 100 µg/mL phenylmethylsulfonyl fluoride, 10 µg/mL DNase I, and 1 mM β-mercaptoethanol and lysed by three passages through a French press at 16,000 psi. After the removal of cell debris by centrifugation at 30,000 × g for 30 min, the supernatant was applied onto a 5 mL HisTrap HP column (GE Healthcare) equilibrated with buffer B3 (50 mM NaH$_2$PO$_4$, 300 mM NaCl, 20 mM imidazole, adjusted to pH 8.0 with NaOH). The column was washed with 10 column volumes (CV) of the same buffer, and protein was eluted at a flow rate of 1 mL/min with a linear imidazole gradient obtained by mixing buffers B3 and B4 (50 mM NaH$_2$PO$_4$, 300 mM NaCl, 250 mM imidazole, adjusted to pH 8.0 with NaOH). Fractions containing high concentrations of BacA-His$_6$ were pooled and dialyzed against 3 L of buffer B5 (20 mM Tris/HCl pH 8.0, 10 mM NaCl, 1 mM β-mercaptoethanol) at 4 °C. After the removal of precipitates by centrifugation at 30,000 × g for 30 min, the protein solution was aliquoted and snap-frozen in liquid nitrogen. Aliquots were stored at –80 °C until further use.

To purify LmdC, the protein was first produced as a His$_6$-SUMO-LmdC fusion (*Marblestone et al., 2006*). To this end, *E. coli* Rosetta(DE3)pLysS cells carrying plasmid pLY015 were grown at 37 °C in 3 L of LB medium supplemented with ampicillin and chloramphenicol. At an OD$_{600}$ of 0.6, the culture was chilled to 18 °C, and protein synthesis was induced by the addition of 1 mM IPTG prior to overnight incubation at 18 °C. The cultures were harvested by centrifugation at 10,000 × g for 20 min at 4 °C and washed with lysis buffer (50 mM Tris/HCl pH 8.0, 300 mM NaCl). Cells were resuspended in lysis buffer supplemented with 5 mM imidazole, 10 mg/mL DNase I, and 100 mg/mL PMSF. After three passages through a French press (16,000 psi), the cell lysate was clarified by centrifugation (30,000 × g, 30 min, 4 °C). The fusion protein was then purified using zinc-affinity chromatography using a 1 mL Zn-NTA column (Cube Biotech) equilibrated with lysis buffer containing 5 mM imidazole. Protein was eluted with a linear gradient of 5–250 mM imidazole in lysis buffer at a flow rate of 1 mL/min. Fractions containing high concentrations of His$_6$-SUMO-LmdC were pooled and dialyzed against 3 L of low-salt lysis buffer (20 mM Tris/HCl pH 7.6, 50 mM NaCl, 10% (v/v) glycerol). After the addition of Ulp1 protease (*Marblestone et al., 2006*) and dithiothreitol (1 mM), the protein was incubated for 4 hr at 4 °C to cleave off the His$_6$-SUMO tag. The solution was centrifuged for 30 min at 38,000 × g and 4 °C to remove precipitates and then subjected to ion exchange chromatography using a 1 mL HiTrap Q HP column (Cytiva, USA) equilibrated with low-salt lysis buffer. His$_6$-SUMO passed the column in the flow-through, and LmdC was eluted with a linear gradient of 150–1000 mM NaCl in low-salt buffer. Fractions containing LmdC were concentrated in an Amicon Ultra-4 10 K spin concentrator (MWCO 10,000; Merck, Germany). After the removal of precipitates by centrifugation at 30,000×g for 30 min, LmdC was further purified by size exclusion chromatography (SEC) on a HighLoad 16/60 Superdex 200 pg column (GE Healthcare, USA) equilibrated with SEC buffer (20 mM Tris/HCl pH 7.4, 150 mM NaCl). Fractions containing pure protein were pooled and concentrated. After the removal of precipitates by centrifugation at 30,000 × g, the protein solution was snap-frozen in liquid $N_2$ and stored at –80 °C until further use.

## LmdC enzymatic activity assay

To test the enzymatic activity of LmdC, peptidoglycan from *E. coli* D456 (*Edwards and Donachie, 1993*) was mixed with 10 µM LmdC in buffer A (20 mM Hepes/NaOH pH 7.5, 50 mM NaCl, 1 mM ZnCl$_2$) or buffer B (20 mM sodium acetate pH 5.0, 50 mM NaCl, 1 mM ZnCl$_2$) in a final volume of 50 µL and incubated at 37 °C for 16 hr in a thermal shaker set at 900 rpm. A mixture of peptidoglycan in buffer B without LmdC was used as a control. The reactions were stopped by heating at 100 °C for 10 min. The peptidoglycan was then further digested overnight with cellosyl, and the reactions were stopped by heating at 100 °C for 10 min. After centrifugation of the samples at 14,000 rpm for 10 min, the supernatants were recovered, reduced with sodium borohydride, acidified to pH 4.0 – pH 4.5 with dilute 20% phosphoric acid and subjected to HPLC analysis as previously described (*Glauner, 1988*).

## Bio-layer interferometry

Bio-layer interferometry analyses were conducted using a BLItz system equipped with High Precision Streptavidin (SAX) Biosensors (Sartorius, Germany). As a ligand, a custom-synthetized N-terminally biotinylated peptide comprising residues Met1 to Gln38 of LmdC (GenScript, USA) was immobilized on the biosensors. After the establishment of a stable baseline, association reactions were monitored at various analyte concentrations. At the end of each binding step, the sensor was transferred into an analyte-free buffer to follow the dissociation kinetics. The extent of non-specific binding was assessed by monitoring the interaction of analyte with unmodified sensors. All analyses were performed in BLItz binding buffer (25 mM HEPES/KOH pH 7.6, 100 mM KCl, 10 mM MgSO$_4$, 1 mM DTT, 10 mM BSA, 0.01% Tween).

## Bioinformatic analysis

Protein similarity searches were performed with BLAST (*Altschul et al., 1990*), using the BLAST server of the National Institutes of Health (https://blast.ncbi.nlm.nih.gov/Blast.cgi). Transmembrane helices were predicted with DeepTMHMM (*Hallgren et al., 2022*), coiled-coil regions with PCOILS (*Lupas, 1996*). The positions of conserved functional domains were determined using the PFAM server (*Mistry et al., 2021*). AlphaFold2 (*Jumper et al., 2021*) and AlphaFold-Multimer (*Evans et al., 2022*), as implemented in the AlphaFold.ipynb notebook on Google Colab, were used to predict the tertiary or quaternary structure of proteins, respectively. SuperPlots (*Lord et al., 2020*) were used to visualize cell length distributions and to evaluate the statistical significance of differences between multiple distributions, employing the PlotsOfData web app (*Postma and Goedhart, 2019*).

## Statistics

All experiments were performed at least twice independently with similar results. No data were excluded from the analyses. Statistical analyses were performed using Excel 2019 (Microsoft), unless indicated otherwise. To quantify imaging data, multiple images were analyzed per condition. The analyses included all cells in the images or, in the case of high cell densities, all cells in a square portion of the images. The selection of the images and fields of cells analyzed was performed randomly.

## Material availability

All biological materials generated in this study are available from the corresponding author upon request.

# Acknowledgements

We thank Julia Rosum for excellent technical assistance, Svenja Urban and Ying Liu for help with the construction of plasmids, Daniela Vollmer for the purification of peptidoglycan, Maria Perez-Burgos and Lotte Søgaard-Andersen for help with the transmission electron microscopy analyses, and Daniela Kiekebusch und Sabine Rosskopf for helpful discussions.

# Additional information

## Funding

| Funder | Grant reference number | Author |
| --- | --- | --- |
| University of Marburg | Core funding | Martin Thanbichler |
| Max-Planck-Gesellschaft | Max Planck Fellowship | Martin Thanbichler |
| Deutsche Forschungsgemeinschaft | 450420164 | Martin Thanbichler |
| Deutsche Forschungsgemeinschaft | 26942323 - TRR 174 | Peter L Graumann |
| Biotechnology and Biological Sciences Research Council | BB/W013630/1 | Waldemar Vollmer |
| Max-Planck-Gesellschaft | PhD fellowship of the IMPRS for Environmental, Cellular and Molecular Microbiology | Manuel Osorio-Valeriano |

The funders had no role in study design, data collection and interpretation, or the decision to submit the work for publication. Open access funding provided by Max Planck Society.

## Author contributions

Sebastian Pöhl, Conceptualization, Formal analysis, Supervision, Validation, Investigation, Visualization, Writing – original draft, Writing – review and editing, Constructed plasmids and strains, performed immunoblot and growth analyses and conducted subcellular localization studies (H. neptunium and R. rubrum); Manuel Osorio-Valeriano, Conceptualization, Formal analysis, Supervision, Investigation, Visualization, Writing – original draft, Writing – review and editing, Constructed plasmids and strains, performed immunoblot analyses and subcellular localization studies, purified proteins and conducted biochemical interaction studies (H. neptunium); Emöke Cserti, Formal analysis, Investigation, Constructed plasmids and strains, performed immunoblot analyses and conducted subcellular localization studies (H. neptunium); Jannik Harberding, Methodology, Writing – review and editing, Established the CRISPRi system for H. neptunium; Rogelio Hernandez-Tamayo, Formal analysis, Investigation, Visualization, Writing – review and editing, Performed the single-particle tracking analysis; Jacob Biboy, Formal analysis, Investigation, Writing – review and editing, Conducted the peptidoglycan analysis; Patrick Sobetzko, Formal analysis, Investigation, Methodology, Writing – review and editing, Analyzed the co-conservation of BacA and LmdC homologs; Waldemar Vollmer, Formal analysis, Supervision, Funding acquisition, Writing – review and editing, Supervised the peptidoglycan analysis; Peter L Graumann, Supervision, Funding acquisition, Writing – review and editing, Supervised the single-molecule studies; Martin Thanbichler, Conceptualization, Formal analysis, Supervision, Funding acquisition, Investigation, Visualization, Writing – original draft, Project administration, Writing – review and editing

## Author ORCIDs

Jannik Harberding ⓘ https://orcid.org/0000-0002-5048-116X
Rogelio Hernandez-Tamayo ⓘ https://orcid.org/0000-0003-4666-3929
Jacob Biboy ⓘ https://orcid.org/0000-0002-1286-6851
Waldemar Vollmer ⓘ https://orcid.org/0000-0003-0408-8567
Martin Thanbichler ⓘ https://orcid.org/0000-0002-1303-1442

Reviewer #1 (Public Review): https://doi.org/10.7554/eLife.86577.3.sa1
Reviewer #2 (Public Review): https://doi.org/10.7554/eLife.86577.3.sa2
Author response https://doi.org/10.7554/eLife.86577.3.sa3

## Additional files

### Supplementary files

• Supplementary file 1. Muropeptide composition of peptidoglycan from wild-type and bactofilin-deficient *H. neptunium* cells. The spreadsheet gives the abundance of different muropeptide species in strains LE670 (wild-type), EC28 (Δ*bacA*), EC23 (Δ*bacD*), and EC33 (Δ*bacAD*), as obtained in the analysis described in *Figure 2—figure supplement 2*.

• Supplementary file 2. Muropeptide composition of peptidoglycan from wild-type and mutant *R. rubrum* cells. The spreadsheet gives the abundance of different muropeptide species in strains S1 (wild-type), SP70 (Δ*bacA$_{Rs}$*), and SP68 (Δ*lmdC$_{Rs}$*), as obtained in the analysis described in *Figure 10—figure supplement 1B*.

• Supplementary file 3. Plasmids, strains, and oligonucleotides were used in this study.

• MDAR checklist

• Source data 1. This file contains source data for *Figure 1F*, *Figure 9G and H*, *Figure 10D*, *Figure 1—figure supplement 2D*, *Figure 9—figure supplement 1C*, *Figure 10—figure supplement 2*.

### Data availability

All relevant data generated in this study are included in the manuscript, the supplemental information, and the source data files.

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

# Appendix 1

**Appendix 1—key resources table**

| Reagent type (species) or resource | Designation | Source or reference | Identifiers | Additional information |
|---|---|---|---|---|
| Gene (*Hyphomonas neptunium*) | *bacA* | GenBank | ABI77372 | |
| Gene (*Hyphomonas neptunium*) | *bacD* | GenBank | ABI76590 | |
| Gene (*Hyphomonas neptunium*) | *lmdC* | GenBank | ABI76766 | |
| Gene (*Rhodospirillum rubrum*) | *bacA* | GenBank | ABC22667 | |
| Gene (*Rhodospirillum rubrum*) | *lmdC* | GenBank | ABC22668 | |
| Strain (*Hyphomonas neptunium*) | LE670 | *Leifson, 1964* | ATCC 15444 | Wild-type strain |
| Strains (*Hyphomonas neptunium*) | LE670 derivatives | This paper | See *Supplementary file 1* | |
| Strain (*Rhodospirillum rubrum*) | S1 | *Molisch, 1907* | ATCC 11170 | Wild-type strain |
| Strains (*Rhodospirillum rubrum*) | S1 derivatives | This paper | See *Supplementary file 1* | |
| Strain (*Escherichia coli*) | BL21(DE3) | Invitrogen | Cat. #: CMC0016 | *E. coli* B *dcm ompT hsdS*(rB- mB-) *gal* |
| Strain (*Escherichia coli*) | Rosetta(DE3)pLysS | Merck Millipore | Cat. #: 70956 | F- *ompT hsdSB*(rB- mB-) *gal dcm* (DE3) pLysSRARE (Cam$^R$) |
| Strain (*Escherichia coli*) | TOP10 | Thermo Fisher Scientific | Cat. #: C404003 | F- *mcrA* Δ(*mrr-hsdRMS-mcrBC*) Φ80*lacZ*ΔM15 Δ*lacX74 recA1 araD*139 Δ(*ara-leu*) 7697 *galU galK rpsL* (Str$^R$) *endA1 nupG* |
| Strain (*Escherichia coli*) | WM3064 | Metcalf (unpublished) | | *thrB1004 pro thi rpsL hsdS lacZ*ΔM15 RP4–1360 Δ(*araBAD*)567 Δ*dapA*1341::[*erm pir*(wt)] |
| Antibody | anti-mNeongreen (Mouse monoclonal) | Chromotek | Cat. #: 32f6 | 1:1,000 |
| Antibody | anti-GFP (Rabbit polyclonal) | Sigma | Cat. #: G1544 | 1:10,000 |
| Antibody | anti-mCherry (Rabbit polyclonal) | BioVision | Cat. #: 5993 | 1:10,000 |
| Antibody | anti-HA (Mouse monoclonal) | Millipore | Cat. #: 05–904 | 1:1,000 |
| Recombinant DNA reagent | pCCFPC-3 (plasmid) | *Jung et al., 2015* | | Integrating plasmid for the generation of C-terminal CFP fusions under the control of P$_{Cu}$, Rif$^R$ |
| Recombinant DNA reagent | pCCFPC-3 derivatives | This paper | See *Supplementary file 1* | |
| Recombinant DNA reagent | pCCHYC-2 (plasmid) | *Jung et al., 2015* | | Integrating plasmid for generation of C-terminal mCherry fusions under control of P$_{Cu}$, Kan$^R$ |
| Recombinant DNA reagent | pCCHYC-2 derivatives | This paper | See *Supplementary file 1* | |

*Appendix 1 Continued on next page*

*Appendix 1 Continued*

| Reagent type (species) or resource | Designation | Source or reference | Identifiers | Additional information |
|---|---|---|---|---|
| Recombinant DNA reagent | pCCHYC-3 (plasmid) | *Jung et al., 2015* | | Integrating plasmid for the generation of C-terminal mCherry fusions under the control of P$_{Cu}$, Rif$^R$ |
| Recombinant DNA reagent | pCCHYC-3 derivatives | This paper | See *Supplementary file 1* | |
| Recombinant DNA reagent | pCCHYN-2 (plasmid) | *Jung et al., 2015* | | Integrating plasmid for the generation of N-terminal mCherry fusions under the control of P$_{Cu}$, Kan$^R$ |
| Recombinant DNA reagent | pCCHYN-2 derivatives | This paper | See *Supplementary file 1* | |
| Recombinant DNA reagent | pCVENC-3 (plasmid) | *Jung et al., 2015* | | Integrating plasmid for the generation of C-terminal Venus fusions under the control of P$_{Cu}$, Rif$^R$ |
| Recombinant DNA reagent | pCVENC-3 derivatives | This paper | See *Supplementary file 1* | |
| Recombinant DNA reagent | pdCas9-humanized (plasmid) | *Qi et al., 2013* | Addgene plasmid #: 44246; RRID: Addgene_44246 | Plasmid carrying a codon-optimized version of the *S. pyogenes dCas9* gene |
| Recombinant DNA reagent | pdCas9-humanized derivatives | This paper | See *Supplementary file 1* | |
| Recombinant DNA reagent | pET21a(+) | Novagen | | Plasmid for the overexpression of C-terminally His$_6$-tagged proteins, Amp$^R$ |
| Recombinant DNA reagent | pET21a(+) derivatives | This paper | See *Supplementary file 1* | |
| Recombinant DNA reagent | pNPTS138 (plasmid) | R. K. Alley, unpublished | | *sacB*-containing suicide vector used for double homologous recombination, Kan$^R$ |
| Recombinant DNA reagent | pNPTS138 derivatives | This paper | See *Supplementary file 1* | |
| Recombinant DNA reagent | pRSFDuet-1 (plasmid) | Novagen | | Plasmid for the coexpression of genes under the control of the T7 promoter |
| Recombinant DNA reagent | pRSFDuet-1 derivatives | This paper | See *Supplementary file 1* | |
| Recombinant DNA reagent | pRXMCS-2 (plasmid) | *Thanbichler et al., 2007* | | Low-copy replicative plasmid for the ectopic expression of genes under the control of P$_{xyl}$, Kan$^R$ |
| Recombinant DNA reagent | pRXMCS-2 derivatives | This paper | See *Supplementary file 1* | |
| Recombinant DNA reagent | pTB146 (plasmid) | Bernhard, unpublished | | Plasmid for overexpression of N-terminally His6-SUMO-tagged proteins, Amp$^R$ |
| Recombinant DNA reagent | pTB146 derivatives | This paper | See *Supplementary file 1* | |
| Recombinant DNA reagent | pXYFPC-2 (plasmid) | *Thanbichler et al., 2007* | | Integrating plasmid for generation of C-terminal eYFP fusions under control of P$_{xyl}$, Kan$^R$ |
| Recombinant DNA reagent | pXYFPC-2 derivatives | This paper | See *Supplementary file 1* | |
| Recombinant DNA reagent | pZVENC-2 (plasmid) | *Jung et al., 2015* | | Integrating plasmid for the generation of C-terminal Venus fusions under the control of P$_{Zn}$, Kan$^R$ |
| Recombinant DNA reagent | pZVENC-2 derivatives | This paper | See *Supplementary file 1* | |
| Sequence-based reagent | DNA oligonucleotides | This paper | See *Supplementary file 1* | |

*Appendix 1 Continued on next page*

*Appendix 1 Continued*

| Reagent type (species) or resource | Designation | Source or reference | Identifiers | Additional information |
|---|---|---|---|---|
| Peptide, recombinant protein | LmdC$_{Hne\ (aa\ 1-38)}$ | GenScript | | Biotin-MAKWSANLKATFDRAFPERQIYHRSGGTVRYISISPWQ |
| Chemical compound, drug | 2,6-diaminopimelic acid (DAP) | Sigma | Cat. #: 33240–5 g | |
| Chemical compound, drug | ampicillin | Carl Roth | Cat. #: K029.3 | |
| Chemical compound, drug | cefalexin | Sigma | Cat. #: C4895-5G | |
| Chemical compound, drug | $CuSO_4$ | Carl Roth | Cat. #: P024.2 | |
| Chemical compound, drug | kanamycin | Carl Roth | Cat. #: T832.3 | |
| Chemical compound, drug | rifampicin | Sigma | Cat. #: R3501 | |
| Chemical compound, drug | $ZnSO_4$ | Carl Roth | Cat. #: K301.1 | |
| Chemical compound | hydroxy coumarin-carbonyl-amino-D-alanine (HADA) | *Kuru et al., 2015* | | Pulse-labeling of sites of ongoing cell wall biosynthesis |
| Software, algorithm | BLAST | *Altschul et al., 1990* | https://blast.ncbi.nlm.nih.gov/Blast.cgi | |
| Software, algorithm | DeepTMHMM | *Hallgren et al., 2022* | https://dtu.biolib.com/DeepTMHMM | |
| Software, algorithm | PCOILS | *Lupas, 1996* | https://toolkit.tuebingen.mpg.de/tools/pcoils | |
| Software, algorithm | PFAM | *Mistry et al., 2021* | http://pfam.xfam.org | |
| Software, algorithm | AlphaFold2 | *Jumper et al., 2021* | https://colab.research.google.com/github/sokrypton/ColabFold/blob/main/AlphaFold2.ipynb | |
| Software, algorithm | AlphaFold-Multimer | *Evans et al., 2022* | https://colab.research.google.com/github/sokrypton/ColabFold/blob/main/AlphaFold2.ipynb | |
| Software, algorithm | SuperPlotsOfData | *Goedhart, 2021* | https://huygens.science.uva.nl/SuperPlotsOfData | |
| Software, algorithm | ImageJ | *Schneider et al., 2012* | https://imagej.net/ij/download.html | Plug-ins: JACoP (*Bolte and Cordelières, 2006*; https://imagej.net/plugins/jacop), MicrobeJ (*Ducret et al., 2016*; https://www.microbej.com/download-2) |
| Software, algorithm | BacStalk | *Hartmann et al., 2020* | https://drescherlab.org/data/bacstalk/ | |
| Software, algorithm | SMTracker | *Oviedo-Bocanegra et al., 2021* | https://sourceforge.net/projects/singlemoleculetracker/ | |
| Software, algorithm | Cell-profiles R package | *Cameron et al., 2014* | https://github.com/ta-cameron/Cell-Profiles | |

