## [Editor Report · eLife assessment]

The manuscript explores the interplay between cytoskeletal bactofilins and cell wall hydrolases in bacterial morphogenesis, utilizing a range of methodologies from bacteriological to biochemical. The study provides **important** insights into the control of peptidoglycan biosynthesis by bactofilin polymers and the function of LdmC, supported by a comprehensive array of genetic, bioinformatic, biochemical, and biophysical tools. These **convincing** findings propose a conserved module governing bacterial morphogenesis, emphasizing the direct association of cell wall remodeling enzymes with a dynamic cytoskeleton, akin to mechanisms observed in other cellular processes such as cell growth and division.

---

## [Referee Report · Reviewer #1 (Public Review)]

In their study, Osorio-Valeriano and colleagues seek to understand how bacterial-specific polymerizing proteins called bactofilins contribute to morphogenesis. They do this primarily in the stalked budding bacterium *Hyphomonas neptunium*, with supporting work in a spiral-shaped bacterium, Rhodospirillum rubrum. Overall the study incorporates bacterial genetics and physiology, imaging, and biochemistry to explore the function of bactofilins and cell wall hydrolases that are frequently encoded together within an operon. They demonstrate an important, but not essential, function for BacA in morphogenesis of *H. neptunium*. Using biochemistry and imaging, they show that BacA can polymerize and that its localization in cells is dynamic and cell-cycle regulated. They further demonstrate that BacA likely limits movement of the elongasome into the stalk, spatially confining its activity. The authors then focus on lmdC, which encodes a putative M23 endopeptidase upstream of bacA in *H. neptunium*, and find that is essential for viability. The purified LmdC C-terminal domain could cleave *E. coli* peptidoglycan in vitro suggesting that it is a DD-endopeptidase. LmdC interacts directly with BacA in vitro and co-localizes with BacA in cells. To expand their observations, the authors then explore a related endopeptidase/bactofilin pair in R. rubrum; those observations support a function for LmdC and BacA in *R. rubrum* morphogenesis as well.

An overall strength of this study is the breadth and completeness of approaches used to assess bactofilin and endopeptidase function in cells and in vitro. The authors establish a clear function for BacA in morphogenesis in two bacterial systems, and demonstrate a physical relationship between BacA and the cell wall hydrolase LmdC that may be broadly conserved. The eventual model the authors favor for BacA regulation of morphogenesis in H. neptunium is that it serves as a diffusion barrier and limits movement of morphogenetic machinery like the elongasome into the elongating stalk and/or bud.

The data presented illuminate aspects of bacterial morphogenesis and the physical and functional relationship between polymerizing proteins and cell wall enzymes in bacteria, a recurring theme in bacterial cell biology with a variety of underlying mechanisms. Bactofilins in particular are relatively recently discovered and any new insights into their functions and mechanisms of action are valuable. The findings presented here are likely to interest those studying bacterial morphogenesis, peptidoglycan, and cytoskeletal function.

---

## [Referee Report · Reviewer #2 (Public Review)]

This is an excellent study. It starts with the identification of two bactofilins in H. neptunium, a demonstration of their important role for the determination of cell shape and discovery of an associated endopeptidase to provide a convincing model for how these two classes of proteins interact to control cell shape. This model is backed up by a quantitative characterisation of their properties using high-resolution imaging and image analysis methods.

Overall, all evidence is very convincing and I do not have many recommendations on how to improve the manuscript.

In my opinion, there are only two issues that I have with the paper:

1. The single particle dynamics of BacA is presented and analysed and I would like to give some suggestions on how to maybe extract even more information from the already acquired data:

1.1. Presentation: Figure 5A is only showing projections of single particle time-lapse movies. To convince the reader that it was indeed possible to detect single molecules it would be helpful if the authors present individual snapshots and intensity traces. In case of single molecules these will show step wise bleaching

1.2. Analysis: Figure 5B and Supplement Figure 1 are showing the single particle tracking results, revealing that there are two populations of BacA-YFP in the cell. However, this data does not show if individual BacA particles transition between these two populations or not. A more detailed analysis of the existing data, where one can try to identify confinement events in single particle trajectories could be very revealing and help to understand the behaviour of BacA in more detail.

2. The title of Fig. 3 says that BacA and BacD copolymerise, however, the data presented to confirm this conclusions is actually rather weak. First, the Alphafold prediction does not show the co-polymer, and second, the in vitro polymerisation experiments were only done with BacA in the absence of BacD. Accordingly, the only evidence that supports this is their colocalization in fluorescence microscopy. I suggest to either weaken the statement or change the title and add more evidence.

Finally, did the authors think about biochemical experiments to study the interaction between the cytoplasmic part of LmdC and the bactofilins? These could further support their model.

---

## [Author Response]

The following is the authors’ response to the original reviews.

**Response to reviewers**

We thank the two reviewers for their constructive criticism, which helped to significantly improve our manuscript.

During the revision process, we had to realize that the localization pattern reported for H. neptunium LmdCN-mCherry was an artifact caused by bleed-through of the BacA-YFP signal in the mCherry channel. More detailed studies showed that the fusion protein was detectable by Western blot analysis but, for unknown reasons, did not produce any fluorescence signal. Therefore, we have now removed the localization data shown in previous Figure 8B,C and Figure 8—figure supplement 1.

To provide more evidence for a functional interaction between BacA and LmdC in H. neptunium, we have now established an inducible CRISPR interference system for this species and used it successfully to deplete LmdC (new Figure 9A-F). The loss of LmdC causes morphological defects very similar to those observed for the ΔbacA(D) mutant. In line with the physical interaction of BacA with the cytoplasmic region of LmdC observed in vitro, these findings support the hypothesis that the two proteins act in the same pathway. Consistent with the results obtained in H. neptunium, the absence of BacA leads to the delocalization of LmdC in R. rubrum. Moreover, we now provide in vivo evidence for a critical role of the cytoplasmic region of LmdC in the interaction of this protein with BacA in R. rubrum cells (new Figure 11). Together, these new findings strongly support the model that BacA and LmdC form a conserved morphogenetic module involved in the establishment of complex cell shapes in bacteria.

Please see below for a more detailed explanation of our new results and for our response to the issues raised in the first round of review.

**Reviewer #1 (Public Review)**
In their study, Osorio-Valeriano and colleagues seek to understand how bacterial-specific polymerizing proteins called bactofilins contribute to morphogenesis. They do this primarily in the stalked budding bacterium *Hyphomonas neptunium*, with supporting work in a spiral-shaped bacterium, *Rhodospirillum rubrum*. Overall the study incorporates bacterial genetics and physiology, imaging, and biochemistry to explore the function of bactofilins and cell wall hydrolases that are frequently encoded together within an operon. They demonstrate an important, but not essential, function for BacA in morphogenesis of *H. neptunium*. Using biochemistry and imaging, they show that BacA can polymerize and that its localization in cells is dynamic and cell-cycle regulated. The authors then focus on lmdC, which encodes a putative M23 endopeptidase upstream of bacA in *H. neptunium*, and find that is essential for viability. The purified LmdC C-terminal domain could cleave *E. coli* peptidoglycan in vitro suggesting that it is a DD-endopeptidase. LmdC interacts directly with BacA in vitro and co-localizes with BacA in cells. To expand their observations, the authors then explore a related endopeptidase/ bactofilin pair in *R. rubrum*; those observations support a function for LmdC and BacA in *R. rubrum* morphogenesis as well.An overall strength of this study is the breadth and completeness of approaches used to assess bactofilin and endopeptidase function in cells and in vitro. The authors establish a clear function for BacA in morphogenesis in two bacterial systems, and demonstrate a physical relationship between BacA and the cell wall hydrolase LmdC that may be broadly conserved. The eventual model the authors favor for BacA regulation of morphogenesis in *H. neptunium* is that it serves as a diffusion barrier and limits movement of morphogenetic machinery like the elongasome into the elongating stalk and/or bud. However, there is no data presented here to address that model and the role of LmdC in *H. neptunium* morphogenesis remains unclear.

We hypothesize that BacA establishes a barrier that prevents the movement of elongasome complexes into the stalk, either directly by sterical hindrance and/or indirectly by promoting the formation of an annular region of high positive inner cell curvature that cannot be passed by the elongasome. To test this model, we have now analyzed the localization dynamics of RodZ, a core structural component of the elongasome complex, in wild-type and ΔbacAD cells. We found that wild-type cells show dynamic YFP-RodZ foci whose movement is limited to the mother cell and the nascent bud, with no signal ob-served in the stalk. In ΔbacAD cells, by contrast, the fusion protein is consistently detected in all regions of the cell, including nascent stalks (new Figure 5). These results support the idea that BacA is required to confine the elongasome to the mother cell and bud regions and, thus, set the limits of the different growth zones in H. neptunium. We also attempted to follow the localization dynamics of other elongasome components, such as PBP2, MreC and MreD, but none of the corresponding fluorescent protein fusions was functional.

In the past, we tried intensively to generate conditional mutants of lmdC, but all attempts to place the expression of this gene under the control of the copper- or zinc-inducible promoters available for *H. neptunium* were unsuccessful. To clarify the role of LmdC in H. neptunium morphogenesis, we have now established an inducible CRISPR interference system for this species and managed to block the expression of lmdC using an sgRNA directed against the 5' region of its non-coding strand. We observed that cells lacking LmdC show a phenotype very similar to that of the ΔbacA mutant. Together with the finding that the N-terminal cytoplasmic region of LmdC physically interacts with BacA, this result strongly supports the hypothesis that BacA and LmdC act in the same pathway, forming a complex that ensures proper morphogenesis in *H. neptunium* (new Figure 9).

The data presented illuminate aspects of bacterial morphogenesis and the physical and functional relationship between polymerizing proteins and cell wall enzymes in bacteria, a recurring theme in bacterial cell biology with a variety of underlying mechanisms. Bactofilins in particular are relatively recently discovered and any new insights into their functions and mechanisms of action are valuable. The findings presented here are likely to interest those studying bacterial morphogenesis, peptidoglycan, and cytoskeletal function.
**Reviewer #2 (Public Review):**
This is an excellent study. It starts with the identification of two bactofilins in H. neptunium, a demonstration of their important role for the determination of cell shape and discovery of an associated endopeptidase to provide a convincing model for how these two classes of proteins interact to control cell shape. This model is backed up by a quantitative characterisation of their properties using high-resolution imaging and image analysis methods.Overall, all evidence is very convincing and I do not have many recommendations on how to improve the manuscript.In my opinion, there are only two issues that I have with the paper:1. The single particle dynamics of BacA is presented as analysed and I would like to give some suggestions how to maybe extract even more information from the already acquired data:1.1. Presentation: Figure 5A is only showing projections of single particle time-lapse movies. To convince the reader that it was indeed possible to detect single molecules it would be helpful if the authors present individual snapshots and intensity traces. In case of single molecules these will show step wise bleaching.

We have now added a supplementary video that shows both time series and intensity traces of individual BacA-YFP molecules (Figure 6—Video 1). It verifies the step-wise bleaching of the particles observed and thus shows that we observe the mobility of single molecules. Moreover, we have now included a supplementary figure that shows all trajectories identified within representative cells. This visualization provides a more comprehensive view of our data and further supports the notion that our analysis is based on the detection of single molecules.

1.2. Analysis: Figure 5B and Supplement Figure 1 are showing the single particle tracking results, revealing that there are two populations of BacA-YFP in the cell. However, this data does not show if individual BacA particles transition between these two populations or not. A more detailed analysis of the existing data, where one can try to identify confinement events in single particle trajectories could be very revealing and help to understand the behaviour of BacA in more detail.

We agree that an analysis of the single-molecule traces for transitions between the mobile and static states would help to achieve a more detailed understanding of the polymerization behavior of BacA. We believe that the dynamic formation, reorganization and disappearance of BacA-YFP foci observed by time-lapse analysis (Figure 4) indicates that BacA undergoes reversible polymerization in vivo. A deeper investigation of this aspect is beyond the scope of the present study and will be performed at a later point.

1. The title of Fig. 3 says that BacA and BacD copolymerise, however, the data presented to confirm this conclusion is actually rather weak. First, the Alphafold prediction does not show the co-polymer, and second, the in vitro polymerisation experiments were only done with BacA in the absence of BacD. Accordingly, the only evidence that supports this is their colocalization in fluorescence microscopy. I suggest either weakening the statement or changing the title adds more evidence.

To support the idea that BacA and BacD interact with each other, we have now added images of cells producing BacA-YFP or BacD-CFP individually (new Figure 3—figure supplement 1B,C). The results obtained show that Bac-YFP alone still forms filamentous structures, whereas BacD-CFP condenses into tight foci in the absence of its paralog. However, when produced together with BacA-YFP, the two proteins colocalize into filamentous structures, supporting the notion that they interact with each other. However, we agree that it is unclear whether BacA and BacD copolymerize into mixed protofilaments or whether they form distinct protofilaments that then interact laterally to form larger bundles. We have therefore replaced the term “co-polymerize” with “assemble” in the heading of this section.

Finally, did the authors think about biochemical experiments to study the interaction between the cytoplasmic part of LmdC and the bactofilins? These could further support their model.

We show the interaction between the cytoplasmic region of H. neptunium LmdC and BacA in Figure 9G,H (previously Figure 8D,E). For technical reasons, it was not possible to synthesize a peptide comprising the corresponding region of R. rubrum LmdC, so that our in vitro analysis is limited to the H. neptunium proteins.

To further support the notion that BacA interacts with the cytoplasmic region of LmdC, we have now analyzed the localization behavior of two LmdC variants with amino acid exchanges in the conserved cytoplasmic β-hairpin motif (new Figure 11). Both variants no longer colocalize with BacA and are no longer enriched at the inner cell curve. Interestingly, these exchanges also affect the enrichment of BacA at the inner cell curvature, suggesting that BacA needs to interact with LmdC for proper localization. It is tempting to speculate that BacA polymers have a preferred intrinsic curvature and that the activity of the BacA-LmdC complexes adjusts cell curvature in a manner that facilitates their association with the inner curve.
**Reviewer #1 (Recommendations for The Authors):**
We have the following specific recommendations for the improvement of the manuscript:1. Several places would benefit from additional quantitation of data:a. Figure 1 and supplements: can cell shape be quantified in a more specific way? (e.g. principle component analysis of shape as in https://onlinelibrary.wiley.com/doi/10.1111/mmi.13218). It looks as if BacD production may partially rescue the bacA shape phenotype?

We have made considerable efforts to establish methods to quantify morphological changes and protein localization patterns in Hyphomonas neptunium. Since standard software packages, such as Oufti or MicrobeJ, are not able to reliably detect stalks and, thus, typically identify buds as separate cells, we have developed our own analysis software (BacStalk; Hartmann et al, 2020, Mol Microbiol), that is optimized for the detection of thin cellular extensions. However, while this software works very well with wild-type cells, it also fails to recognize amorphous cells with multiple, ill-defined extensions. Given these problems in cell segmentation, it is currently not possible to use principle component analysis to obtain a robust measure of the morphological defects of bactofilin mutants in H. neptunium.

b. Figures 2-S2b, 7D and 9-S1b - can the area under the peaks be quantified and compared across strains? Visual examination of the spectra makes it difficult to discern differences.

A direct comparison of the peak areas between strains is not possible, because the absolute values depend on the amount of peptidoglycan used in the muropeptide analyses. It is very difficult to precisely quantify peptidoglycan, which makes it challenging to use equal amounts of material from different strains in the reactions. However, the relative proportion of different muropeptide species, as provided in Figure 2—Dataset 1, faithfully reflects the composition of peptidoglycan and can easily compared between strains.

c. Figure 9E,F, 9-S4d - BacA and LmdC localization in R. rubrum is very difficult to assess. It does not look linear/filamentous in most cells and is difficult to tell if it is associated with the inner curvature. Can you quantify the position of the signal along the short axis of the cell to better demonstrate that?

We agree that a better quantification of the distribution of protein along the cell envelope of R. rubrum is required to support the conclusions drawn. To address this issue, we have now used line scans to measure the fluorescence intensities along the inner and outer curve of cells (n=200 per strain) and visualized the data in the form of demographs. The results clearly show an enrichment of BacA and LmdC at the inner curve in wild-type cells and a disruption of this pattern in various mutant backgrounds (new Figures 10F,G,J and 11D,E).

1. Figure 2-S2A. Does ∆bacD grow better than wild-type? It would also be useful to add growth curves of the bacA complemented strains.

In the case of H. neptunium growth curves are often misleading, because cells start to aggregate at the late exponential phase due to abundant EPS formation. The degree of cell aggregation also depends on the morphology of cells, because EPS production is limited to the mother cell body, which makes it challenging to compare morphologically distinct mutant strains. We have now performed growth assays for all H. neptunium deletion and complementation strains used in the study and limited the analysis of doubling times to the early and mid-exponential phase, in which cells do not yet form visible aggregates. The results obtained are now included in the new Figure 1F and Figure 1—figure supplement 2D. They show that the doubling times of the different bactofilin mutants are close to that of the wild-type strain.

1. Figure 4BC: From the demographs provided, BacA and BacD appear to have different localization dynamics. BacD seems to stay at the base of the stalk, nearest the mother cell, whereas BacA migrates towards to bud? Also, "length" is misspelt in the panels.

During the transition to bud formation, we indeed observe that the localization patterns of BacA and BacD are in many cases not fully superimposable, with BacD lagging behind BacA and forming transient additional clusters in the vicinity of the stalk base. Examples are now shown in (Figure 4—figure supplement 4). This effect explains the distinct patterns in the demographs. We have now modified the text accordingly. We have also corrected the spelling of “length” in the figure.

1. Can BacD polymerize on its own? It colocalizes with BacA in *E. coli* but that does not necessarily mean it co-polymerizes.

Please see our response to a similar issue (point 2) raised by Reviewer #1.

1. Lines 263-266. You use *E. coli* PG as a substrate for LmdC in vitro because "peptidoglycan from *H. neptunium* shows only a low degree of cross-linkage and hardly any pentapeptides." Does this not have relevance to the physiological significance of the observed activity? Or do you presume that LmdC activity (and/or that of other endopeptidases) is very high in *H. neptunium* so it is difficult to detect additional activity using HnPG as a substrate? It would be useful to clarify this logic in the text.

DD-crosslinks are formed by all major peptidoglycan biosynthetic complexes, including the elongasome and the divisome, so that their general relevance to cell growth in *H. neptunium* is beyond doubt. The low degree of crosslinkage observed suggests that *H. neptunium* contains high endopeptidase activity, which cleaves crosslinks after their formation by DD-transpeptidases. We have now added the explanation “likely due to a high level of autolytic activity” to make this point clearer. Whether LmdC makes a major contribution to the low level of crosslinkage remains to be determined. However, our data suggest that it mostly acts in complex with BacA, so that it may only cleave peptidoglycan locally and not have a global effect global on cell wall composition. It would not possible to detect the DD-endopeptidase activity of LmdC using *H. neptunium* peptidoglycan as a substrate, because it has a low content of DD-linked peptide chains. To facilitate the in vitro activity assay, we therefore used highly crosslinked peptidoglycan from a mutant *E. coli* strain.

1. Lines 268-269: Is there some explanation for why monomers do not increase on LmdC treatment? Here quantitation of peaks before and after treatment would allow the reader to more precisely interpret these data.

The absolute peak sizes are not comparable, because there is some variation in the amount of peptidoglycan included in the assays (see also our comments on point 1b raised by Reviewer #1) and the integrated peak areas (which correspond to the amounts of muropeptide species produced) depend on both the height and the width of the peaks, which vary to some degree in different HPLC runs. The relevant measure to compare the muropeptide profiles is therefore the relative content of different muropeptide species in the different conditions. For clarification, we have now added the following sentence to the legend of Figure 8D: “A quantification of the relative abundance of different muropeptide species in each condition, based on a comparison of the relative integrated peak areas, is provided in Figure 8—Dataset 1.” The control reaction lacking LmdC only contains peptidoglycan diluted in buffer and thus provides insight into muropeptide composition of untreated peptidoglycan.

1. Lines 280-283: It would be interesting to know if the transmembrane domain of LmdC is required for its localization since it is dispensable for binding BacA and since LmdC still localizes to foci without BacA.

Given that it is currently not possible to localize LmdC in H. neptunium, we were not able to perform this analysis.

1. Line 296: it is also possible that LmdC localizes with another protein and does not independently assemble into larger complexes.

Since the localization pattern reported for LmdC in the ΔbacAD background is no longer valid, we have not discussed this aspect in the revised version of our manuscript. However, in general, we do not exclude the possibility that LmdC could interact with other peptidoglycan biosynthetic proteins.

1. Line 304-306 and Fig 9: Is the domain organization of RrLmdC the same as for HnLmdC? It would be useful to include its domain organization as well. Also, please add amino acid numbering to Figure 9B.

We have now added a schematic showing the domain organization of LmdC from R. rubrum (new Figure 10B). The protein is highly similar to its homolog from H. neptunium.

1. Line 340-341: "In both cases, they functionally interact with LmdC-type DD-endopeptidases to promote local changes in the pattern of peptidoglycan biosynthesis." This conclusion is not experimentally supported. Since LmdC is essential and you could not make a depletion strain in H. neptunium, it was not shown that the interaction with LmdC is how BacA promotes changes in PG patterning. HADA/FDAA labeling was not performed in R. rubrum, and no global changes in PG chemistry were observed in bacA or lmdC mutants, so you cannot claim BacA or LmdC influences PG patterning there, either. Either soften this statement to a hypothesis or otherwise rephrase.

To further corroborate a functional interaction between BacA and LmdC, we have now established an inducible CRISPRi system to deplete LmdC from H. neptunium cells (see also our comments on the public review of Reviewer #1). We observe that the loss of LmdC leads to a phenotype very similar to that observed for the ΔbacA(D) mutant, supporting the idea that BacA and LmdC act in the same pathway. We have now also performed localization studies of the elongasome component RodZ in H. neptunium, which demonstrate that the spatial distribution of elongasome complexes is affected in the absence of the bactofilin cytoskeleton in H. neptunium. Combined with the observation that LmdC is a catalytically active DD-endopeptidase and its absence leads to morphological defects, these results indicate that BacA, together with LmdC, induces local changes in pattern of peptidoglycan biosynthesis, both by affecting elongasome movement and, likely, by reducing peptidoglycan crosslinking in the cell envelope regions it occupies.

1. Figure 9-S4: there is no panel C (change D to C).

Corrected.

1. Lines 344-355: No data is presented here to support the barrier model of bactofilin function. In addition, it is unclear why cells would take on amorphous shapes instead of extended rod shapes/filaments if elongasome function was not constrained on the longitudinal axis. It would be helpful to have more discussion of the potential mechanisms of LmdC function in H. neptunium in this section of the discussion since that is the emphasis of the results section.

To support the barrier model, we have now compared the localization dynamics of the elongasome component RodZ in wild-type and ΔbacAD cells. The results show that RodZ is excluded from the stalk in the wild-type background, whereas it readily enters the stalk in the mutant cells, leading to the expansion of stalks into large, amorphous extensions. Consistent with these findings, HADA labeling is not observed within the stalks in wild-type cells, whereas it is readily observed in the enlarged stalk structures (pseudohyphae) formed in the mutant cells.

The current model of MreB movement suggests that MreB filaments have an intrinsic curvature and thus preferentially align along regions of similar curvature, which is along the circumference of the cell in rod-shaped geometries. However, previous work has shown that MreB starts to move along randomly oriented trajectories as soon as cells lose their rod-shaped morphology and adopt more spherical shapes (Hussain et al, 2018, eLife). In line with these findings, our current and our previous work (Cserti et al, 2017, Mol Microbiol) indicate that the expansion of the ovoid H. neptunium mother cell prior to the onset of stalk biosynthesis as well as bud formation are mediated by the elongasome complex. Thus, the elongasome can clearly also give rise to shapes other than rods. Interestingly, however, the H. neptunium elongasome also appears to drive the formation of the rod-shaped stalk, possibly by moving around the circumference of the stalk base. Thus, species- or growth phase-dependent regulatory mechanisms or, potentially, differences in the spatial arrangement of the glycan strands within the peptidoglycan layer may result in different modes of elongasome movement and, thus, modulate the morphogenetic activity of elongasome complexes.

1. Lines 395-397: It is also possible that LmdC positioning is dependent on cell morphology, rather than directly on BacA, since morphology is so distorted in bacA mutant cells.

We provide several lines of evidence showing that LmdC and BacA functionally and physically interact (see above), making it highly unlikely that the two proteins are not associated with each other. However, our previous (Figure 10I,J) and new (Figure 11) results suggest that the physical interaction with LmdC and/or or the cell shape-modulating activity of the complex are required for the proper localization of BacA at the inner curve of the cell. This finding may indicate the existence of a self-reinforcing cycle, in which the morphological changes induced by BacA-LmdC assemblies stimulate the recruitment of additional assemblies to their site of action.